# The MIR-NAT *MAPT-AS1* does not regulate Tau expression in human neurons

Rafaela Policarpo[1,2,3], Leen Wolfs[2,3], Saul Martínez-Montero[4], Lina Vandermeulen[1], Ines Royaux[5], Gert Van Peer[6], Pieter Mestdagh[6], Bart De Strooper[2,3,7], Annerieke Sierksma[2,3], Constantin d'Ydewalle[1] *

1 Neuroscience Discovery, Janssen Research & Development, Janssen Pharmaceutica, Beerse, Belgium, 2 Vlaams Instituut voor Biotechnologie-Katholieke Universiteit Leuven Center for Brain & Disease Research, Leuven, Belgium, 3 Laboratory for the Research of Neurodegenerative Diseases, Department of Neurosciences, Leuven Brain Institute, KU Leuven, Leuven, Belgium, 4 Janssen Biopharma, Janssen Research & Development, South San Francisco, United States of America, 5 Discovery Sciences, Janssen Research & Development, Janssen Pharmaceutica, Beerse, Belgium, 6 Biogazelle, Zwijnaarde, Belgium, 7 United Kingdom Dementia Research Institute, University College London, London, United Kingdom

* cvanoutr@its.jnj.com

**Data Availability Statement:** The data generated in this study is included in the paper itself and uploaded as Supporting information files.

## Abstract

The *MAPT* gene encodes Tau protein, a member of the large family of microtubule-associated proteins. Tau forms large insoluble aggregates that are toxic to neurons in several neurological disorders, and neurofibrillary Tau tangles represent a key pathological hallmark of Alzheimer's disease (AD) and other tauopathies. Lowering Tau expression levels constitutes a potential treatment for AD but the mechanisms that regulate Tau expression at the transcriptional or translational level are not well understood. Natural anti-sense transcripts (NATs) are a particular class of long non-coding RNAs (lncRNAs) that can regulate expression of their overlapping protein-coding genes at the epigenetic, transcriptional, or translational level. We and others identified a long non-coding RNA associated with the *MAPT* gene, named *MAPT* antisense 1 (*MAPT-AS1*). We confirmed that *MAPT-AS1* is expressed in neurons in human *post mortem* brain tissue. To study the role of *MAPT-AS1* on *MAPT* expression regulation, we modulated the expression of this lncRNA in human neuroblastoma cell lines and in human induced pluripotent stem cell (iPSC) derived neurons. In contrast to previous reports, we observed no changes on *MAPT* mRNA or Tau protein levels upon modulation of *MAPT-AS1* levels in these cellular models. Our data suggest that *MAPT-AS1* does not regulate Tau expression levels in human neurons *in vitro*. Thus, *MAPT-AS1* does not represent a valuable therapeutic target to lower Tau expression in patients affected by tauopathies including AD.

## Introduction

Alzheimer's disease (AD) is a chronic neurodegenerative brain disorder and accounts for 60–70% of all cases of dementia [1]. It is estimated that someone develops dementia every three seconds, with a predicted annual cost of 1 trillion US dollars. Currently, over 50 million people

**Funding:** This work was supported by VLAIO (R&D grants HBC.2018.2290 and HBC.2020.3236), and European Research Council (ERC) grant CELLPHASE_AD834682 (EU), FWO, KU Leuven, VIB, Stichting Alzheimer Onderzoek, Belgium (SAO), the UCB grant from the Elisabeth Foundation, a Methusalem grant from KU Leuven and the Flemish Government and Dementia Research Institute - MRC (UK). BDS is the Bax-Vanluffelen Chair for Alzheimer's Disease and is supported by the Opening the Future campaign and Mission Lucidity of KUL, Leuven University. There was no additional external funding received for this study.

**Competing interests:** CdY is an employee of Janssen Pharmaceutica, pharmaceutical companies of Johnson&Johnson. In connection with such employment, CdY receives salary, benefits, and stock-based compensations including stock options, restricted stock and other stock-related grants. CdY and SMM hold patents covering methods to modify Tau expression. BDS is scientific founder of Augustine Therapeutics and Muna Therapeutics, two biotech companies that do not work on Tau. This does not alter our adherence to PLOS ONE policies on sharing data and materials.

worldwide live with dementia and the number is expected to rise to 152 million by 2050 [2]. Recently approved drugs that reduce amyloid-β (Aβ) burden have shown limited clinical benefit, emphasising the need to find additional, potentially combinatorial, therapeutic approaches that target other disease features [3].

The presence of extracellular Aβ-containing plaques and the intracellular formation of neurofibrillary tangles (NFTs) composed of hyperphosphorylated and aggregated forms of Tau protein in the brain are neuropathological hallmarks of AD [4]. In humans, Tau protein is encoded by the *MAPT* gene, a 134 kb long gene located on chromosome 17q21, and predominantly expressed in neurons where it plays key roles in microtubule network stabilization, neuronal polarity establishment and signal transduction [5]. Deregulation of Tau protein expression and/or metabolism in the CNS is closely linked to tauopathies, which comprise several disorders mainly characterized by the accumulation of aggregated and hyperphosphorylated Tau protein in the brains of patients, including AD, progressive supranuclear palsy (PSP) corticobasal degeneration (CBD), or Pick's disease (PiD) [6–8]. Additionally, autosomal mutations in *MAPT* cause frontotemporal lobar dementia (FTLD-*MAPT*) [9]. Due to its importance in pathophysiology, the expression *MAPT* at the transcriptional and post-transcriptional level requires tight regulation. Nevertheless, the regulatory mechanisms of *MAPT* expression are not well understood, with most studies focusing on *MAPT* splicing events, the existence of two *MAPT* haplotypes (H1 and H2) and Tau post-translational modifications, and the direct implications of these regulatory elements in neurodegeneration (discussed in [7]).

Long non-coding RNAs (lncRNAs), and particularly natural antisense transcripts (NATs) recently emerged as powerful regulators of gene expression. LncRNAs are defined as transcripts longer than 200 nucleotides with no protein-coding potential [10]. Based on their location within the genome and orientation relative to neighbouring protein-coding genes, lncRNAs are broadly categorized into intergenic, intronic, bidirectional, sense or antisense lncRNAs, and enhancer RNAs [11]. Several reports indicate that the function of lncRNAs is tightly associated to their subcellular localization [12]. The majority of lncRNAs localize to the nuclear compartment, where they mostly regulate gene expression at the epigenetic or transcriptional levels by recruiting chromatin modifiers, acting as molecular decoys, interfering with alternative splicing or influencing mRNA stability. Alternatively, lncRNAs expressed in the cytoplasm can participate in multiple post-transcriptional mechanisms, including regulation of splicing events, mRNA stability and translation efficiency (Reviewed in [11, 13]). Around 40% of all known lncRNAs are specifically enriched in the brain, where they show highly regulated spatiotemporal expression patterns [14], and recent studies have suggested a role for lncRNAs in neurodegenerative and neurodevelopmental disorders [14–20]. The recent identification of multiple NATs at distinct human loci associated with hereditary neurodegenerative disorders, including AD, further demonstrates the potential role of these transcripts in the expression regulation of neurodegeneration-related genes [15].

However, functional studies investigating the role of these NATs in the human brain are still needed. We and others have identified *MAPT-AS1* as a NAT associated with the *MAPT* gene [15, 21–29]. To our knowledge, Coupland et al. were the first attempting to investigate the role of *MAPT-AS1* in the regulation of Tau expression in the context of neurodegeneration, specifically in Parkinson's disease [21]. In this study, *MAPT-AS1* has been proposed to act as a negative regulator of *MAPT* expression at the epigenetic level. In a recent report from Simone and colleagues, the authors used a wider range of models to study the role of *MAPT-AS1* in neurons, including a neuroblastoma cell line (SH-SY5Y), human iPSC-derived motor neurons and an *in vivo* mouse model (hTau) [29]. Simone et al. identified several *MAPT-AS1* transcripts as NATs that overlap with the *MAPT* gene and that contain embedded mammalian-wide interspersed repeat (MIR) motifs. These motifs are complementary to sequences in the 5'

untranslated region (5'UTR) of the *MAPT* mRNA and were proposed to repress Tau translation by competing with ribosomal RNA pairing [29]. Additionally, several studies explored the role of *MAPT-AS1* and its potential use as a prognostic marker and therapeutic target in distinct cancers [22, 24–28]. Here, we confirm that *MAPT-AS1* is expressed in the human brain from both control and AD patients. We also show that modulating the levels of this lncRNA using siRNA-, ASO- and lentiviral-based approaches fails to induce any changes on *MAPT* transcription and/or translation in multiple human cell lines and in human induced pluripotent stem cell (iPSC) derived neurons. Taken together, our data suggest that *MAPT-AS1* does not regulate *MAPT* expression in human neurons *in vitro*.

## Results

### *MAPT-AS1* is a natural antisense transcript (NAT) arising from the *MAPT* gene locus

To investigate whether long non-coding RNAs could regulate *MAPT* expression, we used a variety of bioinformatics approaches to identify putative NATs overlapping with the *MAPT* gene. We used the UCSC Genome Browser (https://genome.ucsc.edu/) and noted the presence of the *MAPT-AS1* locus on the opposite strand and partially overlapping with the first exon of *MAPT*. The genomic locus of *MAPT-AS1* spans approximately 52 kilobases; the mature reference transcript (RefSeq: NR_024559.1) is 840 nucleotides long and contains two exons that do not overlap with any of the *MAPT* exons (Fig 1a and 1b). Simone et al. [29] recently reported on three different transcript variants arising from the *MAPT-AS1* locus: *t-NAT1*, *t-NAT2l* and *t-NAT2s*, where *t-NAT2s* represents the reference sequence (Fig 1a; Table 1). To confirm the identity of the mature transcript, we combined poly(A)+ RNA sequencing data from human brain reference RNA with FANTOM5 capped analysis of gene expression (CAGE) sequencing data to assess read coverage in the *MAPT-AS1* locus (Fig 1c and 1d), and more specifically at the annotated transcription start site (TSS) of *MAPT-AS1*. We also analysed reads from 3'-end RNA-sequencing on human brain reference RNA (Fig 1c and 1e). All three independent but complementary approaches confirmed expression of the reference sequence NR_024559.1, with CAGE-seq and 3'end-seq read coverage coinciding with the annotated TSS and transcript end, respectively (Fig 1c–1e; Table 1). This sequence corresponds to the *t-NAT2s* isoform reported by Simone and colleagues [29]. Additionally, to evaluate the presence of different *MAPT-AS1* isoforms, we performed *de novo* transcript assembly using the StringTie algorithm on internally generated poly(A)+ RNA-sequencing data from human brain reference RNA and we compared the identified isoforms to previously annotated transcripts (Fig 1c, ENST00000579244, ENST00000581125 and ENST00000579599; https://www.ensembl.org/). StringTie identified two *MAPT-AS1* isoforms: STRG2, which is the known ENST00000579244 isoform and a novel *MAPT-AS1* isoform STRG1 (Fig 1c, top panel; Table 1). For the novel isoform, we did not identify a coverage peak overlapping the 3' end. Furthermore, the current data do not support (abundant) expression of the ENST00000581125 and ENST00000579599 isoforms in human brain reference RNA, as suggested by the lack of a CAGE peak for both isoforms. The absence of additional CAGE peaks suggests that there are no other *MAPT-AS1* transcript isoforms transcribed from this locus. We also did not detect any new transcript matching *t-NAT2l*. In summary, our data does not support robust expression of *t-NAT1* nor *t-NAT2l* transcript variants in human brain, while we confirmed the presence of the reference *MAPT-AS1* transcript of 840 nucleotides in length. A coding potential calculator [30] and an open reading frame finder tool (https://www.ncbi.nlm.nih.gov/orffinder/) indicated that *MAPT-AS1* has no protein-coding potential and thus represents a putative NAT.

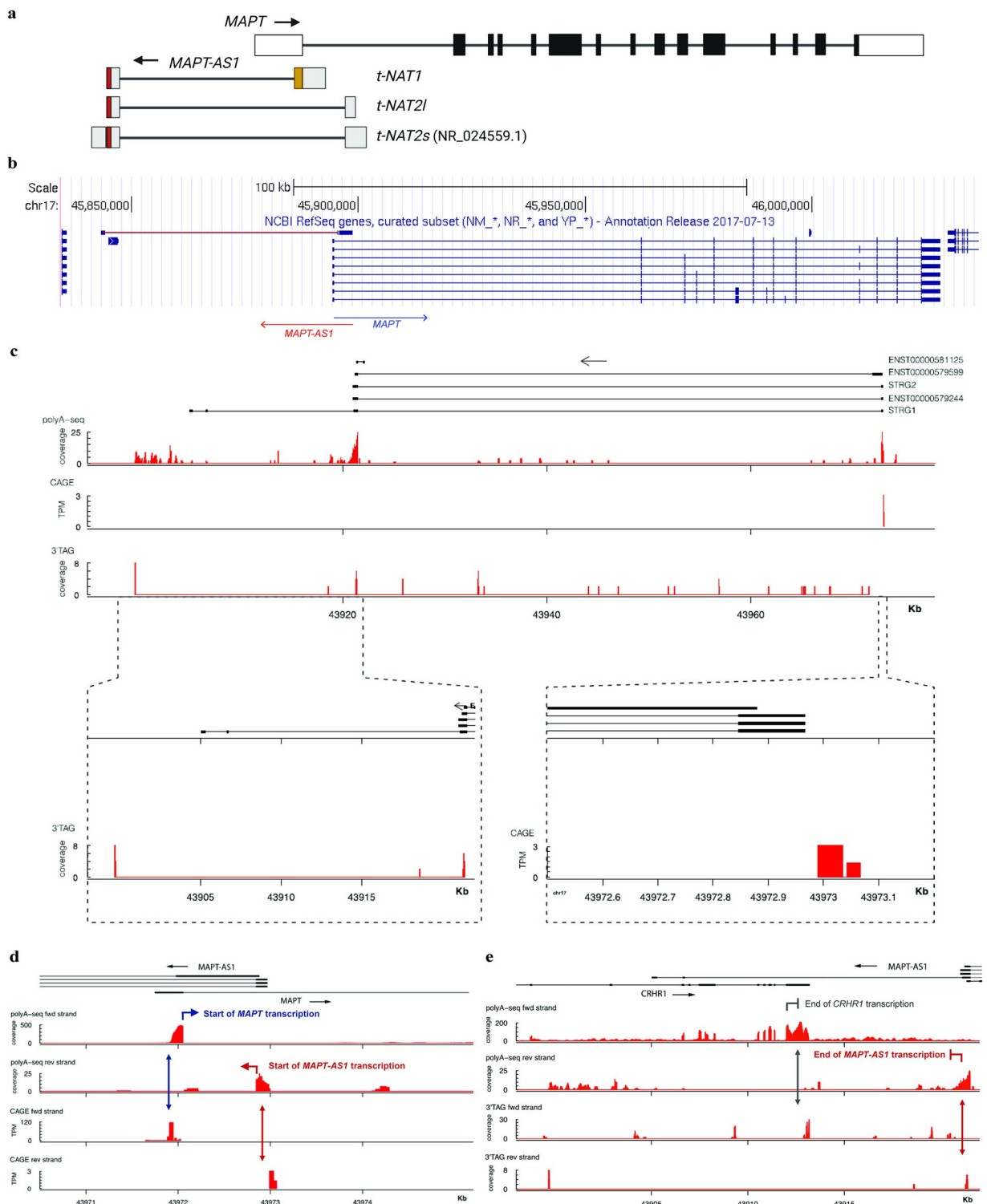

**Fig 1. *MAPT-AS1* is a NAT associated with the *MAPT* gene. a**, Schematic representation of previously reported *MAPT-AS1* isoforms. 5' and 3' UTRs of *MAPT* gene shown as white rectangles; *MAPT* exons shown as black boxes; *MAPT-AS1* exons shown as light grey rectangles; *t-NAT1* sequence partially overlaps with the 5'-UTR of MAPT gene (yellow); MIR elements from *MAPT-AS1* isoforms shown in red. Figure adapted from Simone et al. [29]. **b**, *MAPT-AS1* arises from the antisense strand of the *MAPT* gene in an intronic region downstream of its first exon and does not overlap with any *MAPT* exons. Source: https://genome.ucsc.edu/. **c**, The panels (from top to bottom) display the following information: known and novel *MAPT-AS1* isoforms derived from Ensembl (ENST00000579244, ENST00000581125 and ENST00000579599; source: https://

www.ensembl.org/) and StringTie assembly (STRG1 and STRG2), respectively; internally generated polyA+-sequencing (polyA-seq) coverage data (= read counts; reverse strand) from human brain reference RNA; CAGE data (TPM; reverse strand) from the publicly available FANTOM 5 data set [31]; internally generated QuantSeq 3'-mRNA (= 3'TAG) sequencing coverage data (= read counts; reverse strand) from human brain reference RNA. A zoomed in view for the QuantSeq 3'-mRNA sequencing coverage data at the *MAPT-AS1* 3' end and CAGE TPM at the *MAPT-AS1* transcription start site is also shown. Coordinates are displayed according to the hg19 genome build. **d, e,** Analysis of polyA+ RNA-seq data, CAGE-seq and 3'end-seq data in human brain reference RNA demarking the *MAPT-AS1* transcriptions start (**d**) and end (**e**) site and identifying two potential *MAPT-AS1* isoforms. The same data and annotation as in Fig 1c apply.

In summary, our data demonstrate that *MAPT-AS1* represents a NAT associated to the *MAPT* gene. Multiple sources and methods of RNA sequencing imply that the reference sequence is the predominant transcript isoform expressed in the human brain.

## *MAPT-AS1* is expressed in the human brain

To study the relationship between *MAPT-AS1* and *MAPT* expression, we used publicly available expression data [31, 32] and found that *MAPT-AS1* and *MAPT* are enriched and co-expressed in the brain (Fig 2a–2c). We then measured *MAPT-AS1* expression levels in two different regions of post-mortem brain samples from AD patients and age-matched controls. Consistent with publicly available data, we detected both *MAPT-AS1* and *MAPT* in the entorhinal cortex and hippocampus isolated from controls and AD patients (Fig 2d–2g). *MAPT-AS1* and *MAPT* mRNA levels were reduced in the entorhinal cortex of AD brains as compared to controls (Fig 2d and 2f). The lower expression levels observed for *MAPT-AS1* and *MAPT* are likely the result of neuronal loss as suggested by the reduced expression of two other neuronal genes, *RBFOX3* and *TUBB3*, in both brain regions (S1a–S1d Fig). Expression levels of general neuroinflammatory markers *TREM2* and *GFAP* were overall increased in the same samples, although *TREM2* levels were only significantly increased in the hippocampus. The microglial and astrocytic homeostatic markers *P2RY12* and *ALDH1L1*, respectively, were not differentially expressed between both cohorts (S1e–S1h Fig.).

We also evaluated the expression of the previously identified *t-NAT1*, *t-NAT2l* and *t-NAT2s* transcripts using previously published primer sequences (S1i–S1o Fig) [29]. In line with our RNA sequencing data, *t-NAT2s* which corresponds to the reference *MAPT-AS1* transcript, was the main isoform present in human samples from both hippocampus and entorhinal cortex (S1i, S1j, S1o Fig). However, we found no expression of *t-NAT2l* in either patient group or brain region, and *t-NAT1* was expressed at very low levels and restricted to some individuals (S1k–S1o Fig).

Together, our data indicate that *MAPT-AS1* is expressed in the human brain. In two different AD-relevant brain regions, the reference sequence is the most abundant isoform whereas other reported isoforms [29] are expressed at very low or undetectable levels.

**Table 1. Summary of *MAPT-AS1* isoforms identified in distinct databases or studies.** Isoform names within the same row represent the same transcript and the respective designation according to the source of the identified transcript. Only isoforms discussed in this manuscript were included for comparison.

| Ref Seq | Ensembl | Simone et al. [29] | Policarpo et al. |
|---|---|---|---|
| NR_024559.1 | ENST00000579244 | *t-NAT2s* | STRG2 |
| - | - | - | STRG1 |
| - | ENST00000579599 | *t-NAT2l (?)* | Not confirmed |
| - | - | *t-NAT1* | Not confirmed |
| - | ENST00000581125 | - | Not confirmed |

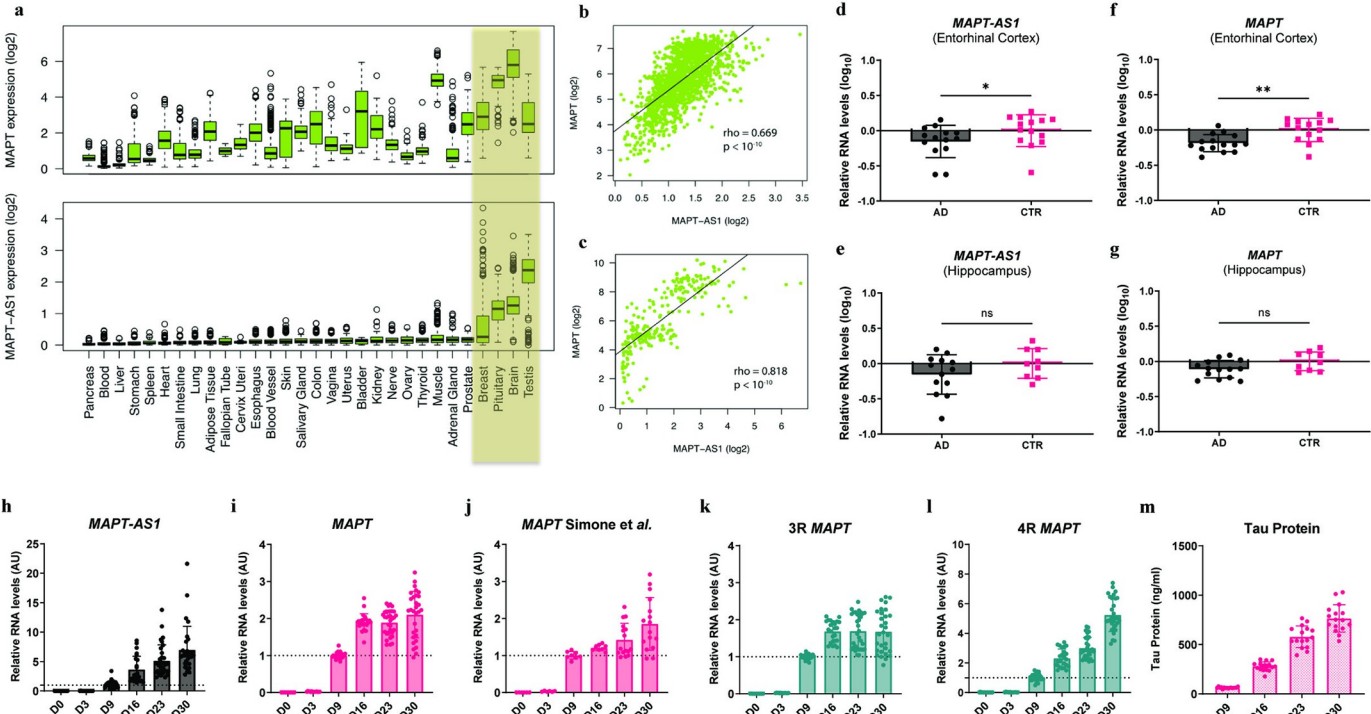

**Fig 2. *MAPT-AS1* is enriched in the brain and dynamically co-expressed with the *MAPT* gene in human neurons. a**, expression distribution of *MAPT* and *MAPT-AS1* across all tissue types in the GTex dataset; $\log_2$ TPM (transcripts per kilobase million) data is shown; tissues ordered based on *MAPT-AS1* expression levels. **b, c,** Expression correlation between *MAPT* and *MAPT-AS1* in human brain samples from GTex (**b**) and FANTOM 5 CAGE datasets (**c**); $\log_2$ of TPM data was used; spearman's *rho* values and *p* values are shown. **d-g,** Expression levels of *MAPT-AS1* (**d, e**) and *MAPT* mRNA (**f, g**) in human brain samples from AD patients and Control individuals evaluated by RT-qPCR; $n$ = 15 AD and 14 CTR for entorhinal cortex; $n$ = 14 AD and 9 CTR for hippocampus; relative RNA level values are normalized to 2 endogenous control genes and calibrated to Control (CTR) group; data are mean ± SD; Mann-Whitney test (**d**) or unpaired t-test (**e-g**) with df = 27 for entorhinal cortex and df = 21 for hippocampus, two-tailed *p* value (*, p ≤ 0.05; **, p ≤ 0.001). **h-l,** Expression levels of *MAPT-AS1* (**h**), *MAPT* mRNA (**i, j**) and specific *MAPT* mRNA isoforms (**k, l**) during differentiation of NGN2-neurons at days 0, 3, 9, 16, 23 and 30 evaluated by RT-qPCR; n = 2 to 6 independent differentiations of NGN2-neurons; relative RNA level values are normalized to 2 endogenous control genes and calibrated to data from 9 days *in vitro*; data are mean ± SD. **m**, Tau protein levels (ng/mL) during neuronal differentiation of NGN2-neurons at days 9, 16, 23 and 30 *in vitro* using a full-length Tau protein MSD assay; n = 2 independent differentiations of NGN2-neurons; data are mean ± SD.

### *MAPT-AS1* is expressed in neuronal models and enriched in the cytoplasm

To further investigate the cell type-specific expression of *MAPT-AS1*, we measured its levels in human induced pluripotent stem cell (iPSC)-derived microglia, astrocytes and neurons. Both *MAPT-AS1* and total *MAPT* levels were particularly enriched in two models of human iPSC-derived neurons compared to astrocytes and microglia where no *MAPT-AS1* expression was detected (S2a Fig). We found a gradual increase in the levels of *MAPT-AS1*, *MAPT* transcripts and Tau protein during neuronal differentiation in a fast maturation inducible model of human iPSC-derived neurons hereby denominated as 'NGN2-neurons' (Fig 2h–2m). Nonetheless, we observed a distinct expression pattern for 3R and 4R Tau-containing transcripts. Specifically, 3R Tau mRNA levels appear to plateau earlier during neuronal maturation, while the increase in 4R Tau isoforms over time suggest a higher degree of maturity of our neuronal cultures [33]. This model has been used by other research groups to investigate Tau modulation approaches [34–36] (S2b Fig). In agreement with our previous human brain data, *MAPT-AS1/t-NAT2s* was the most abundant transcript in NGN2-neurons (S2c and S2f Fig). We did not detect *t-NAT2l* expression in NGN2-neurons, while *t-NAT1* was expressed at low levels in these cells (S2d–S2f Fig). We also observed similar expression patterns of *MAPT-AS1* and

*MAPT* transcripts in a second model of human iPSC-derived cortical neurons generated with a Dual-SMAD inhibition protocol [37–39] (S2g–S2j Fig).

To confirm that *MAPT-AS1* is predominantly expressed in neurons, we assessed its cellular distribution in the brain of AD patients and controls by in situ hybridization. We observed a co-localization of *MAPT-AS1* signal with *MAPT+* or *RBFOX3+* neurons in control and AD brains (S2k Fig). The subcellular localization of long non-coding RNAs dictates largely its potential intracellular functions [12, 40]. *In situ* hybridization on control and AD brains together with RNA fractionation experiments in NGN2-neurons indicated that *MAPT-AS1* localizes predominantly to the cytoplasmic fraction (S2k–S2m Fig). In combination with its genomic location relative to the *MAPT* gene, these observations suggest that *MAPT-AS1* may regulate Tau expression at the post-transcriptional or translational level in the cytoplasm of human neurons.

Taken together, expression profiling of human iPSC-derived cells, RNA fractionation and in situ hybridization experiments indicate that *MAPT-AS1* is expressed predominantly localized in the cytoplasm of human neurons.

## *MAPT-AS1* knockdown with ASOs does not affect Tau expression in neuronal models *in vitro*

*MAPT-AS1* has been recently suggested to repress Tau expression by competing for ribosomal RNA pairing with the *MAPT* mRNA internal ribosome entry site [29]. To independently investigate whether *MAPT-AS1* regulates *MAPT* expression in human neurons, we first utilized antisense oligonucleotides (ASOs) engineered to reduce *MAPT-AS1* levels via an RNase-H dependent mechanism. We screened a total of 42 ASOs tiling the mature reference *MAPT-AS1* transcript in a human neuroblastoma cell line (SK-N-MC) previously reported to express *MAPT-AS1* [21] and identified several candidate lead ASOs that reduced *MAPT-AS1* levels by at least 50% (S3a Fig).

Then, we treated these cells with serially diluted candidate lead *MAPT-AS1* ASOs and identified two lead ASOs (ASO-10 and ASO-16) that dose-dependently reduced *MAPT-AS1* (S3b–S3k Fig). Given that ASO-10 is located within the *MAPT-AS1* hotspot identified in the primary screening and the fact that this ASO showed a slightly higher maximum effect in reducing *MAPT-AS1* levels compared to ASO-16 (64% maximum reduction compared to 51%, respectively), we selected ASO-10 as the lead ASO for targeting *MAPT-AS1* in our experiments (S3e Fig).Treatment with a positive control ASO targeting another long non-coding RNA (*MALAT1*) reduced its target in a dose-dependent manner without affecting *MAPT-AS1*, Tau mRNA or protein levels (S3j–S3mb Fig). Treatment with a non-targeting (scrambled) ASO also did not change *MAPT-AS1* or Tau expression levels at any concentration tested (S3i, S3n, S3o Fig). Cells treated with the most potent *MAPT-AS1* ASO (ASO-10) at various concentrations showed unaltered *MAPT* mRNA and Tau protein levels (S3e, S3p, S3q Fig).

To confirm our findings in differentiated human neurons, we treated NGN2-neurons with several ASOs. We chose a long incubation time (10 days) to span the reported half-life (6–7 days) of Tau protein in human iPSC neurons [41]. We utilized 2 control ASOs. First, a *MALAT1* ASO dose-dependently reduced *MALAT1* levels but did not alter Tau expression at the mRNA or protein level nor *MAPT-AS1* levels (Fig 3a and 3b). Second, treatment with a scrambled ASO did not affect *MAPT-AS1* or Tau expression levels (Fig 3c and 3d). We confirmed that an ASO treatment of 10 days was sufficiently long to reduce Tau expression at the RNA and protein level in a dose-dependent manner using an ASO targeting *MAPT* mRNA (Fig 3e and 3f). Reducing *MAPT* expression did not affect *MAPT-AS1* levels (Fig 3e). Finally,

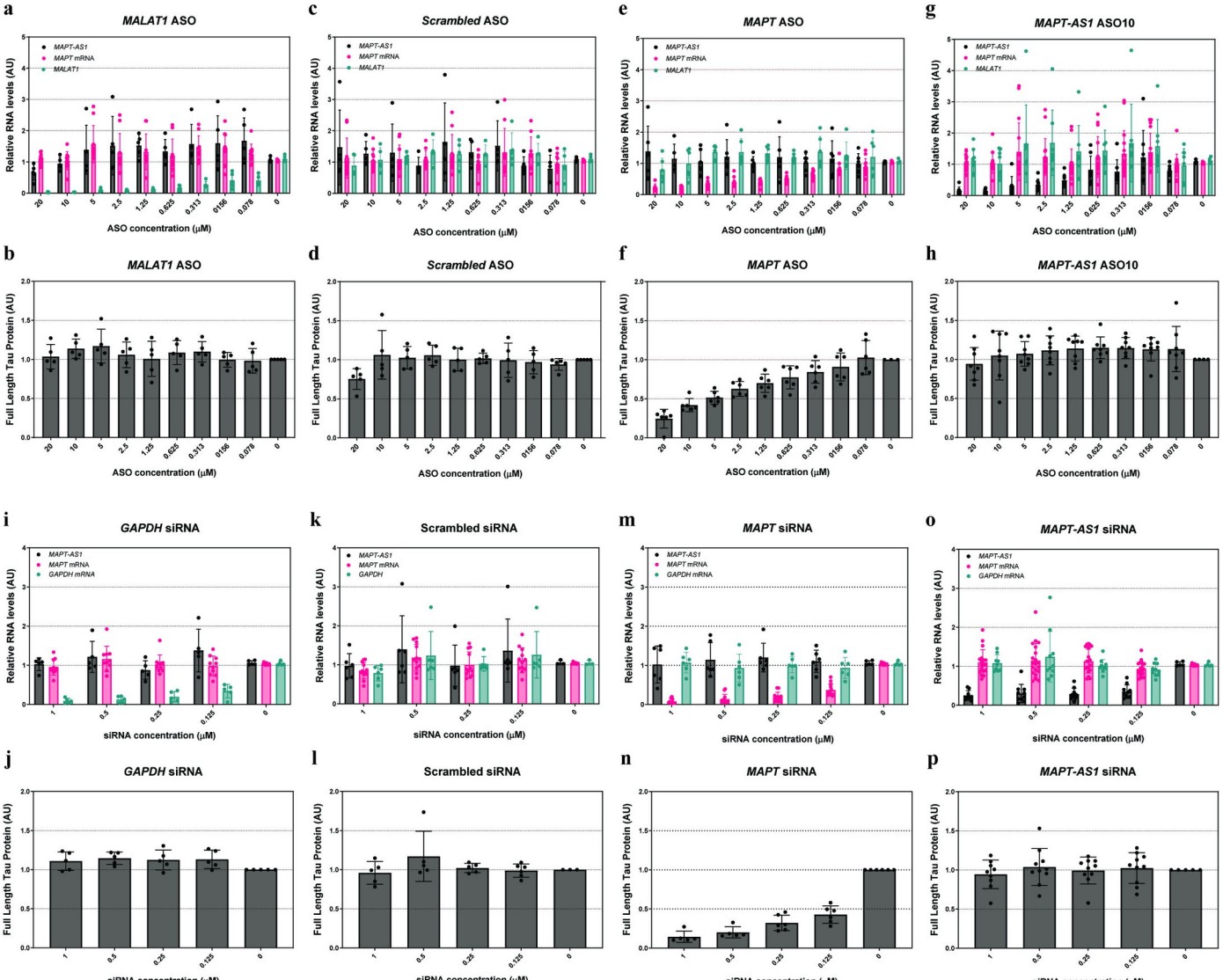

**Fig 3.** *MAPT-AS1* **knockdown does not affect Tau expression at the mRNA or protein level in NGN2-neurons. a-p,** NGN2-neurons were treated with ASOs (**a-h**) or siRNAs (**i-p**) at day 8 and harvested 10 days later at day 18 for RNA and Tau protein analysis. RNA expression levels were evaluated by RT-qPCR; relative RNA level values are normalized to 2 endogenous control genes and calibrated to untreated condition. Tau protein levels were assessed using a full-length Tau protein MSD assay; values scaled to untreated condition (average set to 1); all data are mean ± SD. **a-h,** NGN2-neurons treated with a *MALAT1* (**a, b**), a non-targeting (**c, d**), *MAPT* (**e, f**) or *MAPT-AS1* (**g, h**) ASOs; n = 3–4 independent experiments per ASO. **i-p,** NGN2-neurons treated with a *GAPDH* (**i, j**), a non-targeting (**k, l**), *MAPT* (**m, n**) or *MAPT-AS1* (**o, p**) siRNAs; n = 3–5 independent experiments per siRNA. **a-h,** NGN2-neurons treated with ASOs targeting *MALAT1* (**a, b**), a non-targeting ASO (**c, d**), *MAPT* (**e, f**) or *MAPT-AS1* (**g, h**); n = 3–4 independent experiments per ASO. **i-p,** NGN2-neurons treated with siRNAs targeting *GAPDH* (**i, j**), a non-targeting siRNA (**k, l**), *MAPT* (**m, n**) or *MAPT-AS1* (**o, p**); n = 3–5 independent experiments per siRNA.

treatment with the *MAPT-AS1* ASO dose-dependently reduced *MAPT-AS1* levels but did not change Tau mRNA or protein levels (Fig 3g and 3h).

To rule out that our negative findings were attributed to the use of ASOs, we treated NGN2-neurons neurons with a previously published custom-made short-interference RNA (siRNA) targeting *MAPT-AS1* [21]. A *GAPDH*-targeting positive control siRNA and a non-targeting siRNA did not affect *MAPT-AS1* nor Tau expression levels (Fig 3i–3l). Treatment with an siRNA targeting *MAPT* led to a dose-dependent decrease of *MAPT* mRNA and Tau

protein levels (Fig 3m and 3n). However, siRNA-mediated knockdown of *MAPT-AS1* did not result in altered *MAPT* expression at the RNA or protein level (Fig 3o and 3p).

Next, we evaluated the effect of reducing *MAPT-AS1* transcripts using siRNAs in SH-SY5Y cells. We used both the siRNAs mentioned above, a panel of positive and negative control siRNAs to confirm technical validity of our experiments, as well as the siRNA sequences reported by Simone et al. [29], and their transfection procedures, to ascertain that the discrepancies between our observations and their published findings were not caused by differences in cellular models and/or biochemical assays. Our 2 positive control siRNAs targeting either *GAPDH* or *MAPT* mRNA did reduce their intended target mRNAs, whereas the negative control siRNA did not affect the levels of any RNA investigated in SH-SY5Y cells relative to non-treated cells (S4a and S4b Fig). Targeting *MAPT-AS1* using our siRNA resulted in a robust knockdown of *MAPT-AS1* and *t-NAT2s* transcripts, while it did not affect *MAPT* mRNA levels (S4c and S4d Fig). Treatment with siRNAs reported by Simone et al. [29] resulted in variable knockdown levels of *MAPT-AS1* and its transcript variants (S4e and S4f Fig). In line with our previous observations, we found no evidence for expression of *t-NAT2l* in SH-SY5Y cells and *t-NAT1* was not detected or expressed at variable levels in the same cellular model (S4e Fig).

Taken together, we were unable to confirm robust expression of 2 *MAPT-AS1* transcript variants in the same cellular model and using the same RT-qPCR assays as described by Simone et al. [29]. We were also not able to recapitulate a significant *MAPT-AS1* or *t-NAT* transcripts knockdown in a neuronal cell line using the same siRNAs as originally described [29].

## *MAPT-AS1* overexpression does not affect Tau expression in neuronal models *in vitro*

We also designed lentiviral constructs to overexpress the entire sequence of the reference *MAPT-AS1* transcript either using a neuronal-specific promoter (synapsin-1, SYN1) or a ubiquitously active promoter (cytomegalovirus, CMV). SK-N-MC cells were transduced with up to 30 copies/cell. The two positive control constructs overexpressing eGFP did not change *MAPT-AS1* or Tau expression levels relative to non-treated conditions (S5a–S5d Fig). The presence of GFP signal in cells treated with both SYN1:eGFP and CMV:eGFP indicates that the constructs are active in our model (S5e and S5f Fig). Both lentiviral constructs overexpressed *MAPT-AS1* in a dose-dependent manner in this neuroblastoma cell line (S5g and S5h Fig). However, both *MAPT* mRNA and Tau protein levels remained unchanged (S5i and S5j Fig).

In addition, we generated the same SH-SY5Y cells stably overexpressing *miniNAT* as the minimally required sequence to reduce Tau protein translation as previously reported [29]. We included SH-SY5Y cells stably overexpressing eGFP and *MAPT-AS1* as negative and positive control, respectively. After selection, stably expressing cells showed robust and homogenous eGFP expression confirming similar transduction efficiencies across the different constructs (S6a Fig). We noticed that *MAPT-AS1*, *t-NAT* transcripts and Tau mRNA and protein levels varied with number of passages in SH-SY5Y cells (S6b–S6j Fig). Therefore, to ensure that any potential changes on Tau expression levels are dependent on *MAPT-AS1*, we analyzed both stable and parental SH-SY5Y cell lines at the same passage numbers. We confirmed stable overexpression of both *MAPT-AS1* (also representing its endogenously expressed transcript variants) and *miniNAT* over consecutive passages in SH-SY5Y cells (S6k–S6o Fig). However, stable overexpression of *miniNAT* or *MAPT-AS1* did not show significant changes on *MAPT* mRNA levels (S6p and S6q Fig) and did not result in decreased Tau protein levels compared to untreated conditions (S6r and S6s Fig).

We then transiently overexpressed *MAPT-AS1* in NGN2-neurons. To ensure that potential effects on Tau protein could not be attributed to lentiviral toxicity, we transduced

NGN2-neurons with 1, 3 or 10 copies/ cell. A positive control construct overexpressing eGFP confirmed a high efficiency of transduction in this model (Fig 4a) but it did not change *MAPT-AS1* or *MAPT* expression levels relative to non-treated conditions (Fig 4b and 4c). We also confirmed that both lentiviral constructs overexpressed *MAPT-AS1* in a dose-dependent manner in neurons (Fig 4d and 4f). However, both *MAPT* mRNA and Tau protein levels remained unchanged (Fig 4d–4g).

Finally, we investigated whether transducing NGN2-neurons with a lentivirus overexpressing the *miniNAT* sequence reported by Simone et al. [29] has an effect on Tau expression levels in this model (S7a and S7f Fig). To ensure that the lack of effect on Tau mRNA and protein observed in our previous experiments was not due to a low overexpression of our constructs,

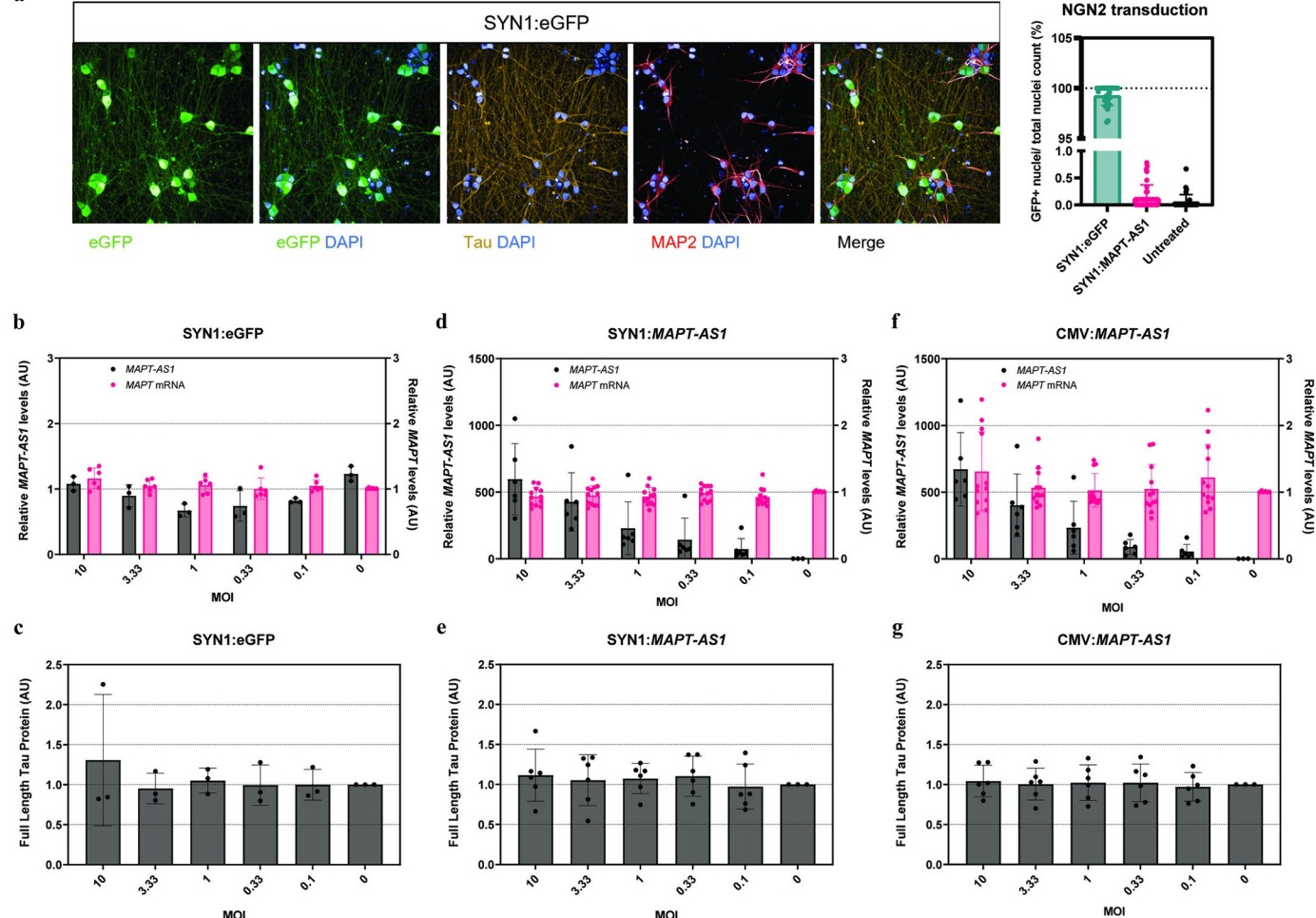

**Fig 4. *MAPT-AS1* overexpression does not affect Tau expression at the mRNA or protein level in NGN2-neurons. a,** left panel: Representative images of NGN2-neurons when transduced with SYN1:eGFP at multiplicity of infection (MOI) of 30, or untreated; scale bar 50 μm. Right panel: % GFP positive cells was calculated using Harmony high-content analysis software; output was based on the formula a/b*100 to calculate the percentage of cells that contain GFP signal (a = number of cells with GFP+ signal, based on Alexa488 channel) inside the nuclei region (b = number of nuclei, based on DAPI channel); One-way ANOVA with Dunnett's multiple comparisons test (****, p < 0.0001); data are mean ± SD. **b-g,** NGN2-neurons were treated with lentiviral constructs overexpressing eGFP (**b, c**) or *MAPT-AS1* (**d-g**) at day 8 and harvested 10 days later at day 18 for RNA and Tau protein analysis; *n* = 3 independent experiments per lentivirus; RNA expression levels were evaluated by RT-qPCR; relative RNA level values are normalized to 2 endogenous control genes and calibrated to untreated condition. Tau protein levels were assessed using a full-length Tau protein MSD assay; values scaled to untreated condition (average set to 1); all data are mean ± SD. Overexpression of the transcripts was controlled by the synapsin-1 (SYN1) or the cytomegalovirus (CMV) promoter. MOI = multiplicity of infection.

this time we increased the number of lentiviral particles up to 30 copies/cell. We confirmed that both SYN1:*miniNAT* and SYN1:*MAPT-AS1* constructs robustly induce overexpression of *miniNAT* and *MAPT-AS1* transcripts, respectively, in an MOI-dependent manner without affecting Tau mRNA levels (S7c and S7e Fig). Additionally, no changes on Tau protein levels were observed upon *miniNAT* overexpression (S7f Fig). Thus, overexpression of *MAPT-AS1* or its previously reported minimally required sequence [29] did not reduce Tau protein translation in human neuronal models.

In summary, we confirmed robust overexpression of *MAPT-AS1* and *miniNAT* using lentiviral-mediated transduction in multiple human cell lines and in human iPSC-derived neurons. However, we found no changes on Tau mRNA nor protein levels when increasing the levels of these *MAPT-AS1* transcripts in our models.

## Discussion

In the present study, we characterized the expression of *MAPT-AS1*, a lncRNA directly associated to the *MAPT* gene, in the human brain tissue of controls and AD patients and in human derived cellular models. We explored whether this lncRNA regulates Tau expression *in vitro*. In line with other data [14, 22, 29], we confirm the neuron-specific, maturation induced and predominantly cytoplasmic expression of *MAPT-AS1*. Although we corroborate previous findings [21, 23, 29, 42] on decreased expression of *MAPT-AS1* and *MAPT* in the entorhinal cortex of AD cases compared to controls, it is most likely that this observation is predominantly driven by the neuronal loss that occurs in AD patients at advanced Braak stages [43].

To unravel whether *MAPT-AS1* would regulate *MAPT*/tau expression in human neurons we used distinct human *in vitro* cellular models, including two neuroblastoma cell lines (SK-N-MC and SH-SY5Y) and a model of Ngn2-induced human iPSC-derived neurons (NGN2-neurons), which have also been used in previous studies [21, 29]. The NGN2-inducible neuronal models have facilitated the production of highly reproducible, mature and functional neurons from diverse types [44, 45] within a short period of time and have already demonstrated their utility to investigate Tau modulation, thereby validating their use to study Tau expression dynamics [34–36]. Using different approaches, we confirmed that NGN2-neurons express neuronal maturation markers at the mRNA and protein levels 2–3 weeks after the initial induction of Ngn2 expression, and a maturation-dependent increase in *MAPT* and *MAPT-AS1* expression levels.

Despite validating our tools, we did not observe changes on Tau mRNA and protein levels upon *MAPT-AS1* knockdown with ASOs, nor when we used a *MAPT-AS1* targeting siRNA sequence previously published by Coupland and colleagues [21]. The same outcome was observed upon overexpression of *MAPT-AS1* with lentiviral constructs in our models. The use of different concentrations and/or MOIs in our study, along with a panel of positive and negative control conditions, allows us to exclude that potential (or lack of) changes in Tau content are derived from either toxicity or from low transfection and/or transduction efficiencies. Indeed, our eGFP-expressing construct under the control of the SYN1 promoter demonstrates high transduction efficiency of NGN2-neurons, thereby ensuring that the increase of *MAPT-AS1* expression using a construct with the same promoter is not restricted to a limited number of cells. In summary, with this set of experiments we were unable to verify that endogenous *MAPT-AS1* controls neuronal Tau levels in human iPSC-derived neurons, as previously described by Simone et al. [29].

In this study, the authors identified three distinct transcripts–*t-NAT2s*, *t-NAT2l* and *t-NAT1* –arising from the *MAPT-AS1* locus which are expressed in different human brain regions [29], where *t-NAT2s* represents the reference *MAPT-AS1* sequence investigated in our study. We acknowledge that isoform-specific effects could potentially explain the discrepancies

observed between our study and the report from Simone et al. [29], with specific isoforms potentially being less responsive to ASO or siRNA treatment and therefore still regulate *MAPT* expression in a sub-stoichiometric manner. However, by using the same primers as Simone et al., we were not able to detect any expression of the *t-NAT2l* isoform in human brain samples or in 2 models of human iPSC-derived neurons, while expression of *t-NAT1* was low and variable across samples. Furthermore, the primers from Simone et al. for the *t-NAT2s* isoform should confirm our observations using our own primers against the *MAPT-AS1* reference sequence, yet they do not recapitulate our observed decrease in *MAPT-AS1* expression in the entorhinal cortex of AD cases [29] (S1i and S1j Fig). A possible explanation for these discrepancies is the fact that we evaluated *MAPT-AS1* levels using a probe-based, exon-spanning assay to account for a high specificity in the detection of this lncRNA. We recognize that the expression (and function) of alternatively spliced lncRNAs might be region- and cell-type specific [46, 47], and that a low abundance of *t-NAT1* and *t-NAT2l* within bulk-tissue RNA or specific brain regions does not preclude their functionality. However, recent RNA sequencing data from the Telomere-to-Telomere (T2T) Consortium provide additional evidence that the *t-NAT2l* isoform is not detected in humans (https://genome.ucsc.edu/) [48]. Simone et al. showed that stable expression of a *miniNAT* sequence in SH-SY5Y retained full capacity to inhibit Tau translation, indicating that this is the minimal sequence necessary for t-NAT transcripts to regulate Tau protein expression [29]. Nonetheless, SH-SY5Y cells stably expressing either *MAPT-AS1* or *miniNAT* did not display lower levels of Tau protein in our hands. Thus, we confirmed that our divergent observations are not due to different cellular models being used and are not mediated by different *MAPT-AS1* isoforms.

Even without the influence of non-coding RNAs, the regulation of Tau expression is complex, not only from a transcriptional and splicing viewpoint, but also from a genomic perspective. The *MAPT* locus contains the largest known block of linkage disequilibrium within the human genome, which spans around ~1.8 Mb and gives rise to two haplotypes denominated as H1 and H2 defined by a 900 kb inversion event encompassing several genes besides *MAPT* [5]. The less common H2 haplotype has been linked to a possible reduction in *MAPT* expression along with a lower risk for developing AD [49]. Similarly, another study found an increase in total *MAPT* expression in the H1/H1 haplotype specifically in the fusiform gyrus, accompanied by a decrease in *MAPT-AS1* RNA levels only in the cerebellar hemisphere of both control and PD patients. This suggests that expression of *MAPT* and *MAPT-AS1* may be regulated in a haplotype-specific manner, independent of disease status [50]. Similarly, Coupland et al. have previously also proposed a haplotype-specific role of *MAPT-AS1* in regulating the expression of Tau splicing variants [21]. Thus, further understanding on the haplotype-specific impact on *MAPT-AS1* expression is warranted.

Apart from haplotype-specific regulation of *MAPT* expression, its 6 different isoforms show distinct expression patterns during brain development [51] and skews in the expression of its 3R to 4R isoform ratios can lead to specific pathologies in adulthood, including CBD, PSP and PiD [8]. Knowing that lncRNAs have the ability to exert their effect during very narrow windows in brain development or only in specific cellular contexts, diseases or pathological states [51–53], we cannot exclude the possibility that our iPSC-based investigation or tissue specific expression did not capture the right developmental or disease stage or cellular context to pick up the regulatory impact of *MAPT-AS1* on *MAPT* expression and/or splicing. Indeed, previous studies have proposed distinct *MAPT-AS1* regulatory mechanisms in different cellular models, suggesting that the regulatory role of this lncRNA might be restricted to highly specific cell types [21, 24–26, 29]. In line with our observations, Bhagat and colleagues have recently reported on 15 lncRNAs with reduced expression in human iPSC-derived neurons carrying *MAPT* mutations, yet *MAPT-AS1* was not among them. Although this strengthens our

conclusion that *MAPT*/*MAPT-AS1* dysregulation is likely not generalizable to all tauopathies [53], we cannot preclude that *MAPT-AS1* expression is affected by other Tau-related disorders [54, 55].

Moreover, its cytoplasmic localization raises the possibility that *MAPT-AS1* plays another role on the regulation of Tau that does not result in total changes on Tau expression levels, or that this regulation happens in a specific subset of neurons. Given the predominant localization of Tau protein in axons, one hypothesis is that *MAPT-AS1* could regulate Tau mRNA transport and/or local translation. Interestingly, Paonessa et al. have found that both missense and splicing FTD-associated *MAPT* mutations drive Tau mislocalization towards MAP2-positive cell bodies and dendrites of cortical neurons, an event that occurs early in FTD pathology [56]. Thus, future experimental approaches will focus on addressing whether *MAPT-AS1* can regulate Tau mRNA transport and/or local translation.

Collectively, our data do not support the biological relevance of *MAPT-AS1* as a negative regulator of Tau protein expression in human neurons. Thus, additional research efforts are needed to explore its biological function(s), including those other than regulating *MAPT*/Tau expression. Several pharmaceutical companies are evaluating safety and efficacy of Tau-targeting therapeutic modalities in clinical trials (Reviewed in [57, 58]). Our data suggest that if *MAPT-AS1* were to have an inhibitory effect on *MAPT*, this appears to be highly specific to particular brain regions and cannot be widely reproduced across cellular models. In this scenario, targeting *MAPT-AS1* as a therapeutic for regulating Tau expression will only work under extremely select conditions, making it less desirable to be pursued. We encourage the scientific community to continue investigating the mechanisms that regulate Tau expression to support the identification and development of additional Tau lowering therapeutics. Ultimately, this will pave the way for the development of novel treatment options for patients affected by AD and other tauopathies.

## Materials and methods

### Bioinformatic analysis

To identify different *MAPT-AS1* isoforms, we performed a transcript-assembly analysis using the StringTie algorithm (v1.3.3b) [59] on internally generated stranded poly(A)+ RNA-sequencing data from human brain reference RNA (ThermoFisher). A library was generated using the TruSeq stranded mRNA library prep kit (Illumina) according to the manufacturer's instructions and sequenced on a NextSeq 500 (Illumina). Reads were mapped to the human reference genome (hg38) using TopHat (v2.1.0) [60]. Transcript assembly was guided by the *MAPT-AS1* reference annotation (Ensembl Release 90). 3'end-sequencing data were generated for from human brain reference RNA (ThermoFisher) using the QuantSeq library prep procedure (Lexogen) according to the manufacturer's instructions and sequenced on a NextSeq 500 (Illumina). Reads were mapped to the human reference genome (hg38) using TopHat.

Expression correlation between *MAPT* and *MAPT-AS1* was evaluated in the GTex RNA sequencing [32] and FANTOM 5 CAGE sequencing [31] datasets. Analyses were performed at the gene level. All analyses were performed using the R statistical programming language.

### Human neuroblastoma cell lines (SK-N-MC and SH-SY5Y)

SK-N-MC (HTB-10) and SH-SY5Y (CRL-2266) cells were obtained from the American Type Culture Collection (ATCC). SK-N-MC cells were maintained in Minimum Essential Medium (MEM; Gibco) supplemented with 10% (v/v) heat-inactivated fetal bovine serum (HI-FBS, Biowest), 1 mM sodium pyruvate (Gibco), 1.5 g/L sodium bicarbonate, 0.1 mM non-essential amino acids (NEEA; Gibco) and 50 µg/ml gentamycin (Gibco). SH-SY5Y cells were

maintained in Dulbecco's Modified Eagle's Medium/Nutrient Mixture F-12 Ham (DMEM/F12; Sigma) supplemented with 10% (v/v) HI-FBS (Biowest), 0.1 mM NEEA (Gibco) and 50 μg/ml gentamycin (Gibco). Cells were kept in a humidified incubator at 37 ˚C and 5% $CO_2$.

## SH-SY5Y stable cell lines

For establishing the stable cell lines (SYN1:*MAPT-AS1*; SYN1:*miniNAT*; SYN1:eGFP), SH-SY5Y cells were seeded in 96-well μclear plates (Greiner Bio-One) at a density of 40,000 cells per well and 24 hours later transduced with 3 custom designed lentiviral constructs, over-expressing, under the neuronal synapsin 1 (SYN1) promoter, the following sequences: 1) the entire cDNA sequence of *MAPT-AS1* (NR_024559.1) except for the PolyA-tail sequence (SYN1:*MAPT-AS1*), 2) the *miniNAT* sequence provided by Simone et al. [29] (SYN1:*mini-NAT*) and 3) a control lentiviral construct overexpressing eGFP (SYN1:eGFP) at a multiplicity of infection (MOI) ratio of 30. MOI = Virus Titer / Number of Cells (S1 File). A total of 6 wells was transduced per lentiviral construct; a total of 18 wells remained untreated (parental control line). The day after transduction, media containing lentiviral particles was replaced by fresh media in each well. On the next day, fresh media containing 1 μg/mL puromycin was added to lentiviral-treated conditions to start selection of stable cell colonies. Cells were cultured for another 5 days with regular media replacement with fresh puromycin containing media until resistant colonies could be identified. Then, cells from 3 independent wells per condition (done twice to expand independent colonies) were split together and transferred into a total of 6 wells per condition in 24-well plates (Falcon). Cells were cultured for another 10 days with regular media replacement with fresh puromycin containing media before being transferred to t25 flasks (two flasks per condition to expand independent colonies). Cells were then plated in 6-well plates at a density of 300,000 cells/well over 3 consecutive passages. Independent lysates from each stable and parental SH-SY5Y cell lines were collected through 3 consecutive passages (#4, #5 and #6 post-thawing, PT). At least 2 wells per condition and per passage were collected for RNA and protein analysis.

## Differentiation of human iPSC-derived microglia

Human induced pluripotent stem cells (hiPSC)-derived microglia were obtained from the ApoEε3/3 hiPSC line (UKBi011-A-3), an isogenic mutant of the ApoEε4/4 genotype parental line (UKBiO011-A) [61]. After thawing the hiPSCs on Matrigel (Corning) coated plates in mTeRS1 medium (Stem Cell Technologies) containing 10 μM ROCK inhibitor (ROCKi, Sigma), the cells were cultured, and the medium changed daily with fresh mTeSR1 medium without ROCKi. When confluency was reached, the cells were passaged with EDTA (Gibco), and differentiated into embryonic bodies (EBs), macrophage precursor factories (MPFs) and microglia using a method described by Sally Cowley et al. [62, 63]. Briefly, hiPSCs were dissociated into single cell suspension and EBs were produced in Aggrewell plates (Stem Cell Technologies) by addition of 0.05 μg/mL BMP-4 (Invitrogen), 0.05 μg/mL VEGF (PeproTech) and 0.02 μg/mL SCF Miltenyi into mTeSR1 medium. After three days of daily medium changes, the EBs were transferred to 6-well factory plates (ThermoFisher) to create MPFs by placing 10–20 EBs per well in X-VIVO15 medium (Lonza) containing 10 U/mL Penicillin-Streptomycin (Gibco), 1% (v/v) GlutaMAX (Gibco), 50 μM 2-Mercaptoethanol (Gibco), 0.1 μg/mL M-CSF (Gibco) and 0.025 μg/mL IL-3 (Gibco). The factories were cultured for about 6–8 weeks with weekly medium changes before the suspension cells produced by the MPFs could be collected for final differentiation into functional and mature microglia. Microglia were cultured at a density of 15,000 cells per well in uncoated 96-well μclear plates (Greiner Bio-One), in Advanced DMEM/F12 medium (Life technologies) containing 10 U/mL Penicillin-

Streptomycin (Gibco), GlutaMAX, 50 µM 2-Mercaptoethanol (Gibco), 0.01 µg/mL GM-CSF (Life Technologies) and 0.1 µg/mL IL-34 (Peprotech) for 14 days, with half medium changes every 2–3 days.

## Differentiation of Human iPSC-derived cortical neurons (Dual-Smad) and human iPSC-derived astrocytes

hiPSCs were differentiated into cortical neuronal progenitor cells (NPCs) by Axol Biosciences (Cambridgeshire, UK), as previously described [38]. Cells were thawed in N2B27 neuronal differentiation medium [1:1 mixture of Neurobasal medium (Gibco) with DMEM:F12 Glutamax (Gibco) with 1% (v/v) B27 supplement (Gibco), 0.5% (v/v) GlutaMAX (Gibco), 0.5% (v/v) N2 supplement (Gibco), 2.5 µg/mL Insulin solution, 25 µM 2-Mercaptoethanol (Gibco), 0.5% (v/v) NEEA (Gibco), 0.5 mM sodium pyruvate (Gibco) and 10 U/mL Penicillin-Streptomycin (Gibco)], supplemented with 20 ng/mL of human recombinant basic fibroblast growth factor (bFGF, Life Technologies).

For neuronal differentiation, cells were dissociated three days after thawing (experimental day 0) using Accutase (Sigma) and seeded on 24-well culture plates (Corning) coated overnight with undiluted Poly-L-ornithine solution (PLO, Sigma) followed by 10 µg/mL of mouse laminin (Sigma), at a density of 125,000 cells per well in neuronal differentiation medium supplemented with 10 µM ROCKi. From day 1 onwards, media changes were done once a week by replacing half of the medium with N2B27 neuronal differentiation media supplemented with 20 ng/mL BDNF (R&D Systems), 20 ng/mL GDNF (R&D Systems), 1 mM db-cAMP (Sigma) and 200 µM L-AA2P (Sigma) until sampling at day 28.

For differentiation of NPCs into astrocytes, cells were dissociated with Accutase (Sigma) three days after thawing (experimental day 3) and seeded on 6-well culture plates (Corning) coated overnight with undiluted PLO (Sigma) followed by 10 µg/mL of mouse laminin (Sigma) at a density of 750,000 cells/well in Astrocyte induction media [N2B27 supplemented with 20 ng/mL of human recombinant CNTF (R&D Systems)]. Between day 12 and 19, medium was switched to Astrocyte Differentiation medium [MEM medium supplemented with 0.6% (v/v) D-glucose (Milipore) and 10% (v/v) HI-FBS] and changed once per week. Astrocytes were cultured for 7–8 weeks and dissociated/sub-plated with Accutase (Sigma) several times until sampling at day 126.

## Differentiation of human iPSC-derived NGN2-inducible neurons (NGN2-neurons)

BIONi010-C-13 gene edited hiPSC cell line was obtained from the European Bank for induced pluripotent Stem Cells (https://cells.ebisc.org/BIONi010-C-13) [34]. Cells were first genotyped for their *MAPT* haplotype and confirmed to be H1/H1 homozygous. To generate NPCs, hiPSCs were thawed and seeded on Matrigel (Corning) coated 6-well plates (Corning) in mTeSR1 medium supplemented with 10 µM ROCKi. One day after seeding, cells were washed once with 1x Dulbecco's Phosphate-Buffered Saline (DPBS, Sigma) and medium was switched to NeuroBasal Medium/NBM [1:1 mixture of DMEM/F12, HEPES (Gibco) with Neurobasal Medium (Gibco), 0.5% (v/v) N2 supplement (Gibco), 0.5% (v/v) B27 supplement without vitamin A (Gibco), 1% (v/v) GlutaMAX (Gibco) and 10 U/mL Penicillin-Streptomycin (Gibco)] supplemented with 2 µg/mL DOX (Doxycycline Hyclate, Sigma) to induce NGN2 expression (day 0). On days 1–2, full medium changes were performed, and DOX was continuously added to the NBM medium at a concentration of 2 µg/mL. On day 3, the resulting immature NGN2-neurons were dissociated with Accutase (Sigma) and seeded on 96-well µclear plates (Greiner Bio-One) or 6-well plates (Corning) coated with Poly-L-ornithine solution (PLO,

Sigma) diluted 1:2 in 1x DPBS overnight followed by a mixture of 5 μg/mL of human (rhLaminin-511, Biolamina) and 10 μg/mL of mouse laminin (Sigma), at a density of 20,000 or 400,000 cells/well, respectively, in NBM medium supplemented with 10 μM ROCKi and 2 μg/mL DOX. One day later, 90% of NBM medium was changed and DOX induction at 2 μg/mL was maintained in NBM medium. On day 5, medium was fully changed to Neuronal Maturation Medium/NMM [NBM as base medium, with additional of 20 ng/mL BDNF (R&D Systems), 10 ng/mL GDNF (R&D Systems), 50 μM db-cAMP (Sigma), 200 μM L-AA2P (Sigma)], and supplemented with 2 μg/mL DOX and 10 μM DAPT (Sigma) to avoid the growth of proliferative cells. After day 5 and until completion of the experiments, half-medium changes were performed twice a week with NMM supplemented with 2 μg/mL DOX until sampling.

## Transfection of neuroblastoma cell lines and transduction of NGN2-neurons

SK-N-MC cells were seeded in uncoated 96-well μclear plates (Greiner Bio-One) at a density of 20,000 cells per well and 24 hours later transfected with serially diluted ASOs (Janssen Biopharma and Axolabs GmbH) by free delivery in the culture medium, starting at 20 μM. SH-SY5Y cells were seeded in uncoated 24-well plates (Falcon) at a density of 150,000 cells per well and 24 hours later transfected with either Accell siRNAs (Dharmacon) by free delivery in the culture medium, or with a final concentration of 2 μM siRNA sequences provided by Simone et al. [29] (Ambion) using RNAiMax (Invitrogen) transfection reagent following manufacturer's instructions. After 48h, cells were collected for RNA and protein analysis.

NGN2-neurons were transfected at DIV8 at a density of 20,000 to 25,000 cells/well with either serially diluted ASOs (Janssen Biopharma/Axolabs) or Accell siRNAs (Dharmacon) by free delivery in the culture medium. Their efficacy in knocking down *MAPT-AS1* levels was evaluated by RT-qPCR. A non-targeting ASO with a non-targeting sequence (Scrambled ASO) was used as negative control. Two ASOs were used as positive controls, one targeting *MALAT1* and another one targeting *MAPT* mRNA. Similarly, one siRNA targeting *GAPDH* and an additional one targeting *MAPT* were used as positive controls for the siRNA-based experiments.

Neurons were transduced with 4 custom designed lentiviral constructs overexpressing the following sequences: 1) the entire cDNA sequence of *MAPT-AS1* (NR_024559.1) except for the PolyA-tail sequence (SYN1:*MAPT-AS1*) under the neuronal synapsin 1 (SYN1) promoter, 2) construct overexpressing the entire cDNA sequence of *MAPT-AS1* (NR_024559.1) except for the PolyA-tail sequence under the cytomegalovirus (CMV) promoter (CMV:*MAPT-AS1*); 3) a control lentiviral construct overexpressing eGFP (SYN1:eGFP) at different MOI ratios; and 4) the *miniNAT* sequence provided by Simone et al. [29] (SYN1:*miniNAT*) under the neuronal synapsin 1 (SYN1) promoter–included in S7 Fig.

Information on lentiviral constructs, siRNAs and ASOs is available at S1 File.

## RNA isolation from human brain samples

Hippocampal and entorhinal cortex tissue samples were obtained from the London Neurodegenerative Diseases Brain Bank and collected in accordance to British legislation and their ethical board [64]. The human study was evaluated and approved by the ethical committees of Leuven University and UZ Leuven [64]. Total RNA from the human brain tissue was extracted by homogenization in TRIzol (Invitrogen) using 1 ml syringes and 22G/26G needles and purified on mirVana spin columns according to the manufacturer's instructions (Ambion). RNA purity (260/280 and 260/230 ratios) and integrity were assessed using Nanodrop ND-1000 (Nanodrop Technologies) and Agilent 2100 Bioanalyzer with High Sensitivity chips (Agilent

Technologies, Inc.) and Qubit 3.0 Fluorometer (Life Technologies), respectively. A total of 300 ng of RNA was used per sample for cDNA conversion and RNA levels evaluated by RT-qPCR. Range of Ct values for each of the transcripts were as follows: no amplification for *t-NAT2l*; 30.9–31.2 for *t-NAT2s*; 34.8–35.2 for *t-NAT1*; 30.5–31.2 for total *t-NAT*; 36.3–37.4 for *MAPT-AS1* (Taqman).

## Multiplex fluorescent *in situ* hybridization (RNAscope)

Human brain samples were obtained from the Netherlands Brain Bank (NBB), Netherlands Institute for Neuroscience, Amsterdam. Written informed consent was given by the donors for brain autopsy and for the use of material and clinical data for research purposes, in compliance with national ethical guidelines [65]. Frozen human brain blocks of superior frontal gyrus from 6 individuals were cryosectioned to a thickness of 10μm using a CryoStar NX70 cryostat (ThermoFisher), layered onto SuperFrost Plus glass slides (ThermoFisher) and further stored at -80˚C before experiments. Sectioned samples on glass slides were processed for in situ hybridization, which was performed using the RNAscope Multiplex Fluorescent V2 Assay (ACD Bio-Techne) according to the manufacturer's instructions. Sections were fixed in cold 4% PFA and dehydrated using a series of ethanol dilution steps, followed by treatment with Hydrogen Peroxide for 10 minutes at RT. Protease digestion using Protease IV provided in the RNAscope kit was carried out for 20 minutes at RT. Hybridization proceeded for 2 hours at 40˚C. The following ACD Bio probes were used: Hs-*RBFOX3*-C1 (415591), Hs-*MAPT-AS1*-C2 (564491-C2), and Hs-*MAPT*-C3 (408991-C3). Hs-*POLR2A*-C1; *PPIB*-C2; *UCB*-C3 were used low-, medium- and high-expressing positive control probes, respectively. Bacterial *DapB* probe was used as a negative control. Slides were stored in 5X SSC buffer overnight at RT. Amplification was carried out the following day using the three amplification reagents provided with the kit. Detection was done with TSA$^®$ Plus Fluorophores FITC (*MAPT-AS1*), Cy5 (*MAPT*) and Cy3 (*RBFOX3*). Slides were incubated for 30 seconds with TrueBlack 1x solution and washed with 1x DPBS followed by DAPI staining. Images were acquired using the Leica SP8× confocal microscope and analysed using the ImageJ software.

## RNA isolation from SK-N-MC, SH-SY5Y and NGN2-neurons

Total RNA from SK-N-MC, SH-SY5Y and NGN2-neurons was extracted using the RNeasy 96 kit or RNeasy Mini Kit (Qiagen) according to manufacturer's recommendations with minor changes. Briefly, cell culture medium was removed and RLT lysis buffer (Qiagen) was added to the cells for 20 minutes at RT with shaking at 500 rpm. No β-mercaptoethanol was added to the RLT lysis buffer. Lysates were either stored at -80˚C and thawed prior to RNA isolation or protocol was carried out immediately as described in the kits. On-column DNase treatment was performed as instructed by the manufacturer. After RNA elution with RNAase-free water, RNeasy columns/ column plates were discarded and the eluate tubes/ plates containing the total RNA were stored at -80˚C until used for cDNA conversion. RNA quantification was performed using the absorbance 260nm value with NanoDrop.

## RNA subcellular fractionation in NGN2-neurons

RNA fractionation was performed using the RNA Subcellular Isolation kit (Active Motif) according to the manufacturer's instructions. Briefly, NGN2-neurons were washed with 1x DPBS dissociated from 6-well plates with Accutase (Sigma). Then, NBM medium was added to each well to detach the cells from the plate and cells were centrifuged at 450 x g for 5 minutes. Cells were washed with 1x DPBS and centrifuged at 14,000 rpm for 5 minutes. To isolate cytoplasmic and nuclear fractions, each pellet was resuspended in 120 μl of Complete

Lysis Buffer and incubated on ice for 10 minutes. Then, cells were centrifuged at 14,000 rpm for 5 minutes at 4˚C. Supernatant (cytoplasmic fraction) was transferred to a new tube without disturbing the pellet (nuclear fraction). Complete Buffer G was added to each fraction as recommended. To extract the RNA fractions, lysates were mixed with 1 volume of 70% ethanol by pipetting up and down 3 times and transferred to the corresponding purification columns. To promote RNA binding to the silica membrane, columns were centrifuged at 14,000 rpm for 1 minute at 4˚C. The eluate was discarded, and columns were washed with Wash Buffer by centrifugation at 14,000 rpm for 1 minute at 4˚C. The eluate was discarded, and the columns were washed with 70% Ethanol using the same centrifugation parameters. The eluate was discarded, and a final centrifugation step was performed at 14,000 rpm for 2 minutes at 4˚C. Then, purification columns were transferred to a new collection tube and 50 µl of RNAse-free water was added to each column. To elute the RNA, columns were centrifuged at 14,000 rpm for 1 minute at 4˚C. Purification columns were discarded and the eluate tubes containing the RNA subcellular fractions were stored at -80˚C until used for cDNA conversion. RNA quantification was performed using the absorbance 260nm value with NanoDrop.

## cDNA conversion and real-time quantitative PCR (RT-qPCR)

Conversion of total RNA to first-strand cDNA was carried out using the High-Capacity cDNA conversion kit in a 20 µl reaction following manufacturer's instructions (ThermoFisher) without RNAse inhibitor in the 2x RT master mix reaction. cDNA samples were stored at -20˚C until analyzed with RT-qPCR.

RT-qPCR experiments were performed in a 10 µl reaction and run for 40 cycles. Prime-Time Gene Expression Master Mix (IDT DNA) or PowerUp SYBR Green Master Mix (ThermoFisher) were used with the appropriate Taqman assays or primers, respectively. RT-qPCR reactions were run in triplicates. RT-qPCR was performed with the QuantStudio™ 7 Flex Real-Time PCR System (Design & Analysis Software v2.3.0; ThermoFisher) or QuantStudio 12K Flex Real-Time PCR System (QuantStudio Flex Software v1.2.4; ThermoFisher). Results were analyzed using qBase+ software v3.2 (Biogazelle) [66]. All expression levels represented as "RNA relative levels" and shown in Y-axis of each figure reflect calibrated normalized relative quantities (CNRQ) [66] unless specified otherwise. CNRQ values are relative quantities that have been normalized to reference genes and calibrated against a control sample. Unlike the ΔΔCt method, which provides a fold-change expression level based on relative Cq values, qBase+ uses an advanced model that accounts for variation in reference genes and allows for multiple reference genes. This gives CNRQ values greater stability and accuracy. In this manuscript, for each experimental design, a panel of 8 different reference genes available in S1 File was evaluated, and their stability was determined using GeNorm analysis [66]. Target gene expression levels were normalized to the 2 most stable reference genes and calibrated to a control sample/group unless specified otherwise. Given the overall low levels of expression of *MAPT-AS1*, as part of our standard operating procedures to measure the levels of this lncRNA in cellular models, we always include the following control conditions in our experimental designs: (1) "no template control": water instead of real sample added to the mix prior to cDNA conversion and (2), "no reverse transcriptase" control i.e. the same sample was analyzed without undergoing cDNA conversion as no reverse transcriptase was added in the mix, and (3) detection of *MAPT-AS1* levels using a probe-based exon-spanning assay.

Information on custom designed primers and IDT DNA pre-designed Taqman RT-qPCR assays and primers is available at S1 File.

## Protein isolation and Tau detection with mesoscale discovery

Protein samples were obtained by lysis in cold RIPA buffer (Sigma) supplemented with phosphatase (PhosStop, Roche) and protease inhibitors (cOmplete, Roche). Lysates were stored at -20˚C prior to the protein analysis.

To quantify Tau protein levels using Mesoscale (MSD) platform, JRD/hTau/24 (same as hTau21, Janssen R&D) [67] capture antibody was diluted in 1x DPBS at a final concentration of 1 µg/mL and added directly into the wells on 96-well sector standard plates (L15XA, MSD). Plates were then sealed and incubated overnight at 4˚C. After overnight incubation, coated plates were blocked with 0.1% (v/v) Blocker Casein (ThermoFisher) in 1x DPBS for 1–2 hours at RT with agitation at 500 rpm. Following this step, plates were washed 5 times with washing buffer [0.05% Tween (v/v) in 1x DPBS]. After washing, standards and samples were added to the plates diluted in RIPA buffer and incubated overnight at 4˚C with shaking at 500 rpm. Human recombinant tau441 protein (TAU-4R-WT, Tebu Bio) was used to generate standard curves and to interpolate Tau protein concentrations. The next day, plates were washed again 5 times as previously described, and the JRD/hTau/43 (Janssen R&D) [68] detection antibody was added at a 1:3000 dilution to the plates and incubated for 2 hours at RT with agitation at 500 rpm. After this step, plates were washed as before, and MSD Read Buffer T (MSD) with surfactant diluted 1:2 in distilled water was added to each well. Plates were immediately read using the MSD SECTOR Imager 6000 (MSD). Tau concentration (ng/mL) was interpolated from human recombinant 2N4R Tau standard curve.

## Western blot analysis

SH-SY5Y protein samples were obtained by lysis in cold RIPA buffer (Sigma) supplemented with phosphatase (PhosStop, Roche) and protease inhibitors (cOmplete, Roche). Lysates were stored at -20˚C prior to the protein analysis. Protein lysates concentrations were measured using a BCA Protein Assay Kit (ThermoFisher). Proteins were separated in 4–12% SDS–polyacrylamide gel (Criterion XT Bis-Tris, Bio-Rad) in MOPS buffer and transferred to 0.2-µm nitrocellulose membrane for 7 min at 2.5A constant (pre-defined Midi gel mixed molecular weight protocol), using the Trans-Blot Turbo Transfer system (Bio-Rad). Immunoblotting of SH-SY5Y was performed with the following primary antibodies: DAKO Polyclonal rabbit anti-human Tau (1:15000; Agilent #A0024) and monoclonal mouse anti-β-actin (1:2000; Sigma #A2228). Secondary antibodies were as follows: donkey-anti-rabbit IgG-HRPO (1:5000; GE Healthcare #NA934V) and goat-anti-mouse IgG-HRPO (1:3000; BioRad #170–6516). Membranes were developed with Amersham Imager 600 (GE Healthcare) and quantified using ImageJ-Fiji (version v1.53c).

## Immunofluorescence

Medium was carefully removed from NGN2-neurons and cells were fixed with 4% (v/v) paraformaldehyde (ThermoFisher) for 15 minutes at RT. Then, neurons were blocked with 5% (v/v) normal goat serum (Sigma) diluted in 1x DPBS supplemented with 0.3% (v/v) Triton X-100 (ThermoFisher) for 1 hour at RT, followed by overnight incubation at 4˚C with primary antibodies against Oct3/4 (1:50; SantaCruz #sc-5279), Sox2 (1:250; Cell Signalling #3579S), Ngn2 (1:500; Millipore #AB5682), Nestin (1:250; Millipore #MAB5326), NeuN (1:1000; Millipore #ABN78), Tuj1 (1:1000; Abcam #AB52623), HT7 (1:250; ThermoFisher MN1000) and Map2 (1:7000; Abcam #AB5392). The following day, cells were washed 3 times for 5 minutes in 1x DPBS, and then incubated for 60 minutes with Alexa-Fluor conjugated secondary antibodies (ThermoFisher: Mouse AF488, #A-11029; Rabbit AF568, #A-11011; Chicken AF647, #A-21449) diluted 1:400 in 1x DPBS supplemented with 0.3% (v/v) Triton X-100 at RT. Images

were acquired using the Opera Phenix Plus High Content Screening System confocal microscope.

## Antisense oligonucleotide synthesis

Antisense oligonucleotides (ASOs) were synthesized as previously described [69] by Janssen Biopharma (South San Francisco, USA) or by Axolabs GmbH (Germany). All ASOs are 20 nucleotides long. The 5 nucleotides on either end contain 2'-O-methoxyethyl (2'MOE) nucleotides whereas the central segment comprises 2'-deoxynucleotides. All cytosine residues are 5'-methylcytosines. All inter-nucleotide linkages are phosphorothioate. The lyophilised ASOs are reconstituted to a stock concentration of 1 mM in 1x DPBS. The *MAPT-AS1* ASOs tiling the predicted mature transcript (NR_024559.1, 840 bp long) with no overlaps were designed without any restrictions.

## Statistical analysis

All data are presented as mean and standard deviation (SD) unless specified otherwise. Statistical significance was set at $\alpha = 0.05$. Statistical analysis was performed using Graphpad Prism (v8.4.2). Normality analysis was always performed prior to choosing the post-hoc test to analyse the data. Statistical analysis of expression data from human brain samples was performed using parametric unpaired t-test for normally distributed groups or nonparametric Mann-Whitney U-test for non-normal data. For multiple comparison datasets, One-Way ANOVA with Dunn's multiple comparisons test was used, unless specified otherwise. More details on each individual statistical analyses can be found in the relevant figure legends.

## Supporting information

**S1 Fig. Expression profile of neuronal and neuroinflammatory markers, and *t-NAT* transcripts in the human brain. a-h**, mRNA expression levels of neuronal markers *RBFOX3* (**a, b**) and *TUBB3* (**c, d**); and reactive astrocytic and microglial markers *GFAP* (**e, f**) and *TREM2* (**g, h**), respectively, in human brain samples from AD patients and Control individuals evaluated by RT-qPCR; n = 15 AD and 14 CTR for entorhinal cortex; n = 14 AD and 9 CTR for hippocampus; relative RNA level values are normalized to 2 endogenous control genes and calibrated to Control (CTR) group; data are mean ± SD; Mann-Whitney test (**a,b**) or unpaired t-test (**c-n**) with df = 27 for entorhinal cortex and df = 21 for hippocampus, two-tailed p-value (*, $p \leq 0.05$; **, $p \leq 0.001$). **i-n**, Expression levels of *t-NAT* transcripts (**i-n**) in human brain samples from AD patients and Control individuals evaluated by RT-qPCR; n = 15 AD and 14 CTR for entorhinal cortex; n = 14 AD and 9 CTR for hippocampus; relative RNA level values are normalized to 2 endogenous control genes and calibrated to Control (CTR) group; data are mean ± SD; datapoints for *t-NAT1* are missing due to undetectable levels of this transcript in some individuals. Mann-Whitney test or unpaired t-test with df = 27 for entorhinal cortex and df = 21 for hippocampus, two-tailed p value (*, $p \leq 0.05$; **, $p \leq 0.001$). o, Cq values from *t-NAT* transcripts in human brain samples were obtained with RT-qPCR. Crossed samples correspond to no amplification. Data are mean.
(TIF)

**S2 Fig. *MAPT-AS1* localizes predominantly to the cytoplasm of human neurons. a,** Expression levels of *MAPT-AS1* and *MAPT* mRNA in SH-SY5Y and SK-N-MC cells, and human iPSC derived models were evaluated by RT-qPCR; *n* = 1–2 samples per cell type; astrocytes and microglia analysed at day 126 and day 14, respectively; NGN2-neurons or human iPSCs differentiated to cortical neurons using the Dual-SMAD at days 30 and 28 *in vitro*, respectively;

relative RNA level values are normalized to 2 endogenous control genes and calibrated to average; data are mean (± SD, when applicable). **b**, Human iPSCs show expression of stem cell markers Sox2 and Oct3/4 at day 0, and NGN2 after treatment with doxycycline for 4 days. Immature neurons at day 9 displayed a neuron-like morphology accompanied by expression of Nestin. Neuronal maturation markers MAP2, Tau, TUJ1 and NeuN are expressed at day 16. DAPI staining was used to detect cell nuclei. Scale bar 50μm; *n* = 4 independent differentiations of NGN2-neurons. **c-e**, Expression levels of *t-NAT* transcripts during differentiation of NGN2-neurons at days 0, 3, 9, 16, 23 and 30 evaluated by RT-qPCR; *n* = 2 to 6 independent differentiations of NGN2-neurons; relative RNA level values are normalized to 2 endogenous control genes and calibrated to data from 9 days *in vitro*; data are mean ± SD. **f**, Cq values from *t-NAT* transcripts in NGN2-neurons samples were obtained with RT-qPCR. Crossed samples correspond to no amplification. Data are mean. **g-h**, Expression levels of *MAPT-AS1* (**g**) and *MAPT* mRNA transcripts (**h-j**) during differentiation of Dual-SMAD neurons at days 7, 14, 21 and 28 evaluated by RT-sqPCR; n = 8 independent wells per timepoint from 1 differentiation of Dual-SMAD neurons; relative RNA level values are normalized to 2 endogenous control genes and calibrated to data from 7 days *in vitro*. Data are mean ± SD. **k**, *In situ* hybridization showing *MAPT* (red), *MAPT-AS1* (green) and *RBFOX3* (white) expression in *post-mortem* brains from CTR and AD patients. Representative images from 1 CTR and 1 AD patient. Scale bar 30μm. **l, m,** Expression levels of *MAPT-AS1*, *MAPT*, *MALAT1* (nuclear control) and *GAPDH* (cytoplasmic control) were evaluated by RT-qPCR in sub-cellular fractions of NGN2-inducible neurons at DIV23; *n* = 2 lysates from 1 differentiation of NGN2-neurons; values scaled to total fraction; data not normalized to reference genes and shown as mean ± SD.
(TIF)

**S3 Fig. *MAPT-AS1* knockdown does not affect Tau expression at the mRNA or protein level in SK-N-MC cells. a,** SK-N-MC cells were treated with 42 ASOs spanning the entire *MAPT-AS1* transcript at a single dose (5 μM) and harvested after 48 hours for RNA analysis; n = 2 independent screening experiments; *MAPT-AS1* RNA expression levels were evaluated by RT-sqPCR using 3 independent assays; relative RNA level values are normalized to 2 endogenous control genes and calibrated to untreated condition; Kruskal-Wallis test with Dunn's multiple comparisons test; data are mean ± SD. **b-k, S**K-N-MC cells were treated with serially diluted (20 to 0.078μM) lead candidate *MAPT-AS1* ASOs (**b-h**, ASO-2, -8, -9, -10, -16, -21, -24), a *MALAT1* ASO (**i, j**) or a non-targeting ASO (**k**) and harvested after 72 hours for RNA analysis. **l-q**, SK-N-MC cells were treated with a *MALAT1* ASO (**l, m**), a non-targeting ASO (**n, o**), or the lead *MAPT-AS1* ASO (**p, q**) and harvested after 72 hours for RNA and Tau protein analysis. *n* = 3–4 independent experiments per ASO; RNA expression levels were evaluated by RT-qPCR; relative RNA level values are normalized to 2 endogenous control genes and calibrated to untreated condition. Tau protein levels were assessed using a full-length Tau protein MSD assay; values scaled to untreated condition (average set to 1); all data are mean ± SD.
(TIF)

**S4 Fig. Transient *MAPT-AS1* siRNA-mediated in SH-SY5Y cells. a-e,** SH-SY5Y cells were treated with different siRNAs and harvested 48 hours later for RNA analysis. Results from siRNA sequences obtained from Simone *et al*. [29] are shown on the right panel of each graph, as mentioned. Expression levels of *GAPDH* (**a**) and *MAPT* mRNA (**b**), *MAPT-AS1* (**c**) and *t-NAT* transcripts (**d-f**) were evaluated by RT-qPCR; *n* = 3 independent experiments per siRNA; data from different experiments indicated as circles, squares or triangles, respectively; relative RNA level values are normalized to 2 endogenous control genes and calibrated to untreated

condition; data are mean ± SD. Kruskal-Wallis test with Dunn's multiple comparisons test; significant differences are indicated in the graphs and compared to the untreated group (*, p ≤ 0.05; **, p ≤ 0.01; ***, p ≤ 0.001).
(TIF)

**S5 Fig. Lentiviral-mediated overexpression of *MAPT-AS1* does not affect Tau expression levels in SK-N-MC cells. (a-j)** SK-N-MC cells were treated with lentiviral constructs overexpressing **(a-f)** eGFP or **(g-j)** *MAPT-AS1* and harvested 48 hours later for RNA and TAU protein analysis; n = 2–4 independent experiments. **(e, f)** eGFP expression was confirmed 48 hours after treatment with both SYN1:eGFP and CMV:eGFP constructs. Representative images 48 hours after treatment at MOI30; scale bar 400 μm. **(a, b, g, h)** RNA expression levels were evaluated by RT-sqPCR; relative RNA level values are normalized to 2 endogenous control genes and calibrated to untreated condition; data are mean ± SD. **(c, d, i, j)** TAU protein levels were assessed using a full-length Tau protein MSD assay; values are scaled to untreated condition (average set to 1); all data are mean ± SD.
(TIF)

**S6 Fig. Stable *MAPT-AS1* or *miniNAT* overexpression does not change Tau expression in SH-SY5Y cells. a,** eGFP expression in SH-SY5Y cells was confirmed after treatment with a SYN1:eGFP construct at a multiplicity of infection (MOI) of 30. Representative images 72 hours after puromycin treatment. Scale bar 400μm. **b-i**, Expression levels of *MAPT* (**b-d**) or *MAPT-AS1* (**e**) and *t-NAT* (**f-i**) transcripts in SH-SY5Y cells at passage 3 or 17 post-thawing (#3 or #17 PT, respectively). RNA expression levels were evaluated by RT-qPCR; *n* = 3 SH-SY5Y lysates per passage; relative RNA level values are normalized to 2 endogenous control genes and calibrated to lowest passage number (#3 PT); data are mean ± SD. **j**, Tau protein levels in SH-SY5Y cells at passage 3, 11 or 17 post-thawing (#3, #11 or #17 PT, respectively) were assessed using Western Blot analysis; *n* = 3 SH-SY5Y lysates per passage; Tau protein levels were normalized to β-actin levels and scaled to the lowest passage number (#3 PT; average set to 1); data are mean ± SD. **k-s**, SH-SY5Y cells were treated with lentiviral constructs at a multiplicity of infection (MOI) of 30 and cells stably expressing eGFP, *MAPT-AS1* or *miniNAT*, with exception for the parental line which remained non-treated. Cell lysates were harvested from 3 consecutive passages for RNA and Tau protein analysis. **k-o**, Expression levels of *miniNAT* (**k**), *MAPT-AS1* (**l**) and *t-NAT* transcripts (**m-o**) were evaluated by RT-qPCR; *n* = 2–3 independent lysates per each condition and per passage; data from different experiments indicated as circles, squares or triangles, respectively; relative RNA level values are normalized to 2 endogenous control genes and calibrated to parental line group; Dunn's multiple comparisons test (*, p ≤ 0.05; **, p ≤ 0.01; ***, p ≤ 0.001; ****, p < 0.0001). All data are mean ± SD. **p, q**, *MAPT* mRNA expression levels in SH-SY5Y stable cell lines were evaluated by RT-qPCR using two independent primer sets; *n* = 2–3 SH-SY5Y lysates per passage; relative RNA level values are normalized to 2 endogenous control genes and calibrated to parental line group; Dunn's multiple comparisons test. **r, s,** Tau protein levels from SH-SY5Y lines were assessed using a full-length Tau protein MSD assay and normalized to total protein (**r**) or Western Blot analysis and normalized to β-actin levels (**s**); *n* = 2–3 independent lysates per each condition and per passage; values scaled to parental line group; One-way ANOVA with Holm-Šídák's multiple comparisons test (*, p ≤ 0.05).
(TIF)

**S7 Fig. *miniNAT* overexpression does not affect Tau expression at the mRNA or protein level in NGN2-neurons.** NGN2-neurons were treated with lentiviral constructs overexpressing eGFP (**a, b**), *MAPT-AS1* (**c, d**) or *miniNAT* constructs (**e, f**) at day 8 and harvested 10

days later at day 18 for RNA and Tau protein analysis; n = 3 independent experiments per lentivirus. RNA expression levels were evaluated by RT-qPCR; relative RNA level values are normalized to 2 endogenous control genes and calibrated to untreated condition. Tau protein levels were assessed using a full-length Tau protein MSD assay; values scaled to untreated condition (average set to 1); all data are mean ± SD.
(TIF)

**S1 File. Customized VectorBuilder lentiviral constructs information.** VectorBuilder Virus ID, Abbreviation, Target RNA/Protein, Viral Type, Promotor and Titer are shown; **Customized and commercially available Accel siRNAs sequences (Dharmacon) and pre-designed and custom-designed Silencer Select siRNAs (Ambion; Simone et *al.* publication).** Full Name, Abbreviation, Reference, Sense and Antisense Sequences are shown. Abbreviation corresponds to the name given in this manuscript; **ASOs sequences.** ASO name, Target Gene, Sequence, Wing Chemistry and Gapmer Chemistry are shown. ASO name corresponds to the name given in this manuscript; **GeNorm human reference genes, custom designed primers and IDT DNA pre-designed Taqman RT-qPCR assays and primers.** For Genorm human reference genes and custom designed primers, gene name, Primer and Sequence are shown. FWD = forward; REV = reverse; IDT DNA pre-designed Taqman RT-qPCR assays and primers can be found at https://eu.idtdna.com/site/order/qpcr/predesignedassay.
(PDF)

## Acknowledgments

We thank the London Neurodegenerative Diseases Brain Banks, the Netherlands Brain Bank, all the individuals and their families for providing human brain tissue samples. We would like to acknowledge Dr. Wei-Ting Chen for helping with the human brain cryosections to perform multiplex fluorescent in situ hybridization. We thank Dr. Lujia Zhou and Dr. Alfredo Cabrera for providing us with lysates from human iPSC derived astrocytes and Dual-Smad neurons.

## Author Contributions

**Conceptualization:** Rafaela Policarpo, Constantin d'Ydewalle.

**Data curation:** Rafaela Policarpo, Gert Van Peer, Pieter Mestdagh, Constantin d'Ydewalle.

**Formal analysis:** Rafaela Policarpo, Gert Van Peer, Pieter Mestdagh.

**Funding acquisition:** Rafaela Policarpo.

**Investigation:** Rafaela Policarpo.

**Methodology:** Rafaela Policarpo, Leen Wolfs, Saul Martínez-Montero, Lina Vandermeulen, Ines Royaux, Gert Van Peer, Pieter Mestdagh.

**Resources:** Saul Martínez-Montero, Ines Royaux, Constantin d'Ydewalle.

**Supervision:** Bart De Strooper, Annerieke Sierksma, Constantin d'Ydewalle.

**Visualization:** Gert Van Peer.

**Writing – original draft:** Rafaela Policarpo, Annerieke Sierksma, Constantin d'Ydewalle.

**Writing – review & editing:** Rafaela Policarpo, Saul Martínez-Montero, Pieter Mestdagh, Bart De Strooper, Annerieke Sierksma, Constantin d'Ydewalle.

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
