## [Decision Letter · Decision Letter 0]

3 Oct 2024

PONE-D-24-29865The MIR-NAT MAPT-AS1 does not regulate Tau expression in human neuronsPLOS ONE

Dear Dr. d’Ydewalle,

Thank you for submitting your manuscript to PLOS ONE. After careful consideration, we feel that it has merit but does not fully meet PLOS ONE’s publication criteria as it currently stands. Therefore, we invite you to submit a revised version of the manuscript that addresses the points raised during the review process.

**As you see from the comments, both reviewers are generally positive and the manuscript was well-accepted. However, both reviewers raised a number of points that need to be thoroughly **
**addressed. Please reduce the number of references to data not shown and increase the number of supplemental figures as necessary.**

We look forward to receiving your revised manuscript.

Kind regards,

Efthimios M. C. Skoulakis, PhD

Academic Editor

PLOS ONE

Journal Requirements:

“This work was supported by VLAIO (R&D grants HBC.2018.2290 and HBC.2020.3236), and European Research Council (ERC) grant CELLPHASE_AD834682 (EU), FWO, KU Leuven, VIB, Stichting

Alzheimer Onderzoek, Belgium (SAO), the UCB grant from the Elisabeth Foundation, a Methusalem grant from KU Leuven and the Flemish Government and Dementia Research Institute - MRC (UK). BDS is the Bax-Vanluffelen Chair for Alzheimer’s Disease and is supported by the Opening the Future campaign and Mission Lucidity of KUL, Leuven University.”

“This work was supported by VLAIO (R&D grants HBC.2018.2290 and HBC.2020.3236), and European Research Council (ERC) grant CELLPHASE_AD834682 (EU), FWO, KU Leuven, VIB, Stichting Alzheimer Onderzoek, Belgium (SAO), the UCB grant from the Elisabeth Foundation, a Methusalem grant from KU Leuven and the Flemish Government and Dementia Research Institute - MRC (UK). BDS is the Bax-Vanluffelen Chair for Alzheimer’s Disease and is supported by the Opening the Future campaign and Mission Lucidity of KUL, Leuven University”

“This work was supported by VLAIO (R&D grants HBC.2018.2290 and HBC.2020.3236), and European Research Council (ERC) grant CELLPHASE_AD834682 (EU), FWO, KU Leuven, VIB, Stichting

Alzheimer Onderzoek, Belgium (SAO), the UCB grant from the Elisabeth Foundation, a Methusalem grant from KU Leuven and the Flemish Government and Dementia Research Institute - MRC (UK). BDS is the Bax-Vanluffelen Chair for Alzheimer’s Disease and is supported by the Opening the Future campaign and Mission Lucidity of KUL, Leuven University.”

4. Thank you for providing the following Funding Statement: 

“CdY is an employee of Janssen Pharmaceutica, pharmaceutical companies of Johnson&Johnson. In connection with such employment, CdY receives salary, benefits, and stock-based compensations including stock options, restricted stock and other stock-related grants. CdY and SMM hold patents covering methods to modify Tau expression. BDS is scientific founder of Augustine Therapeutics and Muna Therapeutics, two biotech companies that do not work on Tau.”

We note that one or more of the authors is affiliated with the funding organization, indicating the funder may have had some role in the design, data collection, analysis or preparation of your manuscript for publication; in other words, the funder played an indirect role through the participation of the co-authors.

If the funding organization did not play a role in the study design, data collection and analysis, decision to publish, or preparation of the manuscript and only provided financial support in the form of authors' salaries and/or research materials, please review your statements relating to the author contributions, and ensure you have specifically and accurately indicated the role(s) that these authors had in your study in the Author Contributions section of the online submission form. Please make any necessary amendments directly within this section of the online submission form.  Please also update your Funding Statement to include the following statement: “The funder provided support in the form of salaries for authors [insert relevant initials], but did not have any additional role in the study design, data collection and analysis, decision to publish, or preparation of the manuscript. The specific roles of these authors are articulated in the ‘author contributions’ section.”

If the funding organization did have an additional role, please state and explain that role within your Funding Statement.

Please also provide an updated Competing Interests Statement declaring this commercial affiliation along with any other relevant declarations relating to employment, consultancy, patents, products in development, or marketed products, etc. 

Additional Editor Comments (if provided):

Reviewers' comments:

Reviewer's Responses to Questions

**Comments to the Author**

1. Is the manuscript technically sound, and do the data support the conclusions?

Reviewer #1: Yes

Reviewer #2: Yes

2. Has the statistical analysis been performed appropriately and rigorously? 

Reviewer #1: Yes

Reviewer #2: Yes

3. Have the authors made all data underlying the findings in their manuscript fully available?

Reviewer #1: No

Reviewer #2: Yes

4. Is the manuscript presented in an intelligible fashion and written in standard English?

Reviewer #1: Yes

Reviewer #2: Yes

5. Review Comments to the Author

Reviewer #1: The manuscript submitted by d’Ydewalle explores one of the many mechanisms by which tau protein levels are regulated in physiology and pathology that is mediated by natural antisense transcripts or NATs, a subclass of lncRNA. In particular, they focused on a previously described NAT, MAPT-AS1, which has been described before and shown to regulate MAPT expression.

Using different neuronal models such as neuroblastoma cell lines, iPSC-derived neurons and human-derived samples, the authors explore variations in MAPT mRNA levels and tau protein levels after silencing or over-expressing MAPT-AS1.

The main observation made is that, despite previously published work by Simone et al. (2021) showing that MAPT-AS1 acts as a translational repressor of MAPT, the authors didn’t find a significant alteration in MAPT mRNA or protein levels after modulating MAPT-AS1 expression.

Although there could be several technical/methodological reasons to, at least partially, explain the disparities in the observations, the submitted work is well-designed, analyzed and presented. The results are well described and the discussion section elaborates on many of the reasons that could explain the ambiguity of the results described here versus the ones described by others.

In general, I consider that the manuscript is suitable for publication but I would like to point out some minor comments that raised my attention:

1. Based on the data shown in Figure 1 it is not clear to me why the authors state that they don’t find a robust expression of MAPT-AS1 transcript variants t-NAT1 and t-NAT2l (line 116).

2. What is the authors’ consideration on the potential degradation of RNA that could occur in the human brain samples based on the post-mortem interval (qPCR results of Figure 2d-g).

3. Along the text, I found the use of the term “iNeurons” (to denominate human iPSC-derived neurons) not appropriate. To the best of my knowledge, “iNeurons” refers to neurons obtained after direct conversion of fibroblast, without the need to reprogram the cells into a pluripotent state as is done in iPSC-derived neurons (see Mertens et al., 2021, Cell Stem Cell). In my opinion, the use of the term “iNeurons” in the text is not correct and could lead to misinterpretations by readers.

4. Based on the data shown in Supplementary Fig 2a, RNA expression levels of MAPT-AS1 and MAPT are almost undetectable in the neuroblastoma cell lines, in particular in the SH-SY5Y cell line. However, the authors used these models to explore the functional relationship between MAPT-AS1 and MAPT by modulating MAPT-AS1 RNA levels by ASOs (in SK-N-MC cells; Supplementary Fig. 3) or by siRNAs (in SH-SY5Y cells; Supplementary Fig. 4). If MAPT-AS1 RNA levels are too low/almost undetectable, it is possible that this gene is actually not functional in the two models, so decreasing its RNA levels (either by ASOs or siRNAs) will have no expected impact (as it was observed).

5. In Supplementary Figure 3f you can distinguish two populations of data. For example, are all the data points with the highest tau protein abundance at each ASO concentration coming from the same replicate? Would more replicates help on data interpretation?

6. The authors described the impact of reducing MAPT-AS1 RNA levels on tau mRNA and protein levels in SH-SY5Y cells (lines 235-254) but Supp. Fig. 4 only shows quantification of RNA levels. Tau protein level quantification is needed to rule out the hypothesis that decreasing MAPT-AS1 RNA levels can increase tau protein abundance by increasing tau translational rate (as postulated by Simone et al.).

7. Although the authors raised the point about the incubation time of ASOs and half-life time tau (lines 208-209), and this aspect looks to be solved for iPSC-derived neurons in Fig. 3 (MAPT ASOs and siRNAs cause a decrease in tau protein levels -Fig. 3f and 3n-), this is apparently not contemplated when applying ASOs over SK-N-MC cells (Supp. Fig. 3) or siRNAs over SH-SY5Y cells (Supp. Fig. 4). The authors choose a 10 days incubation time for ASOs and siRNAs over iPSC-derived neurons, but they used only 48h of incubation over neuroblastoma cell lines (lines 482-483). Unless a piece of data shows that 48h is enough to induce a decrease in tau protein levels in the neuroblastoma cell lines by ASOs or siRNAs, two days could be a short period of time (especially if compared versus the 10 days used on the iPSC-derived neurons).

Finally, I would like to explain that I based my negative answer to question #3 (about the availability of data underlying the findings in the manuscript) on the fact that there are some claims in the manuscript based on “data not shown”.

Reviewer #2: 1. The authors integrated multiple RNA-seq datasets in Fig 1c/d. Why didn’t they provide an overview or a comprehensive IGV plot showing the read coverage across the entire MAPT and MAPT-AS1 regions, instead of displaying only a partial view?

2. Did the authors use an endogenous control, such as GAPDH or 18S rRNA, for the qPCR experiments shown in Fig 2d-g? If so, it should be clearly stated in the main text. Additionally, what does the "Relative RNA level" on the Y-axis represent? Given that the data range appears quite narrow, would presenting it on a "log10" scale make the differences more apparent?

3. Could the authors provide a more detailed explanation of the 3R and 4R expression levels, as well as the changes in their ratios, as shown in Fig 2k and 2l?

4. The data variation in Fig 3a-g appears to be quite substantial. Could the authors provide an explanation for this variability?

6. PLOS authors have the option to publish the peer review history of their article (what does this mean?). If published, this will include your full peer review and any attached files.

Reviewer #1: No

Reviewer #2: No

---

## [Author Response · Author response to Decision Letter 0]

6 Nov 2024

please see the "response to reviewers" file containing a point-by-point reply to the reviewers' comments.

---

## [Decision Letter · Decision Letter 1]

20 Nov 2024

The MIR-NAT MAPT-AS1 does not regulate Tau expression in human neurons

PONE-D-24-29865R1

Dear Dr. d’Ydewalle,

We’re pleased to inform you that your manuscript has been judged scientifically suitable for publication and will be formally accepted for publication once it meets all outstanding technical requirements.

Kind regards,

Efthimios M. C. Skoulakis, PhD

Academic Editor

PLOS ONE

Additional Editor Comments (optional):

Reviewers' comments:

Reviewer's Responses to Questions

**Comments to the Author**

1. If the authors have adequately addressed your comments raised in a previous round of review and you feel that this manuscript is now acceptable for publication, you may indicate that here to bypass the “Comments to the Author” section, enter your conflict of interest statement in the “Confidential to Editor” section, and submit your "Accept" recommendation.

Reviewer #1: All comments have been addressed

Reviewer #2: (No Response)

2. Is the manuscript technically sound, and do the data support the conclusions?

Reviewer #1: Yes

Reviewer #2: Yes

3. Has the statistical analysis been performed appropriately and rigorously? 

Reviewer #1: Yes

Reviewer #2: Yes

4. Have the authors made all data underlying the findings in their manuscript fully available?

Reviewer #1: Yes

Reviewer #2: Yes

5. Is the manuscript presented in an intelligible fashion and written in standard English?

Reviewer #1: Yes

Reviewer #2: Yes

6. Review Comments to the Author

Reviewer #1: (No Response)

Reviewer #2: (No Response)

7. PLOS authors have the option to publish the peer review history of their article (what does this mean?). If published, this will include your full peer review and any attached files.

Reviewer #1: No

Reviewer #2: No

---

## [Editor Report · Acceptance letter]

10 Dec 2024

PONE-D-24-29865R1 

PLOS ONE

Dear Dr. d’Ydewalle, 

I'm pleased to inform you that your manuscript has been deemed suitable for publication in PLOS ONE. Congratulations! Your manuscript is now being handed over to our production team.

Kind regards, 

on behalf of

Dr. Efthimios M. C. Skoulakis 

Academic Editor

PLOS ONE